# The Impact of Mode of Birth, and Episiotomy, on Postpartum Sexual Function in the Medium- and Longer-Term: An Integrative Systematic Review

**DOI:** 10.3390/ijerph20075252

**Published:** 2023-03-24

**Authors:** Anne-Marie Fanshawe, Ank De Jonge, Nicole Ginter, Lea Takács, Hannah G. Dahlen, Morris A. Swertz, Lilian L. Peters

**Affiliations:** 1Department of General Practice & Elderly Care Medicine, University Medical Center Groningen, University of Groningen, P.O. Box 196, 9700 AD Groningen, The Netherlands; 2Midwifery Academy Amsterdam Groningen, InHolland, 9713 GL Groningen, The Netherlands; 3Midwifery Science, Amsterdam UMC Location Vrije Universiteit Amsterdam, De Boelelaan 1117, 1081 HZ Amsterdam, The Netherlands; 4Department of Genetics, University Medical Center Groningen, University of Groningen, 9700 AD Groningen, The Netherlands; 5School of Nursing and Midwifery, Western Sydney University, Blacktown, NSW 2148, Australia; 6Department of Psychology, Faculty of Arts, Charles University, 128 08 Prague, Czech Republic

**Keywords:** delivery, obstetric, pregnancy, sexual behaviour, sexual health, physiological, postpartum period, mothers, midwifery

## Abstract

(1) Background: Sexual function can be affected up to and beyond 18 months postpartum, with some studies suggesting that spontaneous vaginal birth results in less sexual dysfunction. This review examined the impact of mode of birth on sexual function in the medium- (≥6 months and <12 months postpartum) and longer-term (≥12 months postpartum). (2) Methods: Literature published after January 2000 were identified in PubMed, Embase and CINAHL. Studies that compared at least two modes of birth and used valid sexual function measures were included. Systematic reviews, unpublished articles, protocols and articles not written in English were excluded. Quality was assessed using the Newcastle Ottawa Scale. (3) Results: In the medium-term, assisted vaginal birth and vaginal birth with episiotomy were associated with worse sexual function, compared to caesarean section. In the longer-term, assisted vaginal birth was associated with worse sexual function, compared with spontaneous vaginal birth and caesarean section; and planned caesarean section was associated with worse sexual function in several domains, compared to spontaneous vaginal birth. (4) Conclusions: Sexual function, in the medium- and longer-term, can be affected by mode of birth. Women should be encouraged to seek support should their sexual function be affected after birth.

## 1. Introduction

Birth can have profound effects on women’s lives, including on their sexual function in the longer-term. Decreased sexual functioning is an important indicator of maternal morbidity [1], and determinant of quality of life in women of reproductive age; sexual dysfunction rates were found to be up to 64% at six months postpartum, and that approximately 6.4% of women experienced no physical pleasure in their sexual relationships at 18 months postpartum [2,3,4].

The American Psychiatric Association defines sexual function as “a person’s ability to respond sexually or to experience sexual pleasure” [5]. Up until the publication of DSM-V, women’s and men’s sexual responses were thought to follow the same linear pattern (desire, arousal, pattern) [6]. It is now recognised that different genders differ in their sexual function [7]. Additionally, it is recognised that what is considered ‘normal’ sexual function varies widely among women because of the complex nature of women’s sexuality, which can be affected by psychological, physical and cultural factors [8].

An important aspect that may impact sexual function is mode of birth, which can be categorised into spontaneous vaginal birth, vaginal birth with an episiotomy, assisted vaginal birth, planned caesarean section or emergency caesarean section. Data from the Euro-Peristat study (2016) indicated that 66.2% of all births in the EU were spontaneous vaginal births, 7.5% were assisted vaginal births, 10.7% were planned caesarean sections, and 12.9% were emergency caesarean sections [9,10]. Additionally, the rate of episiotomy ranged from 16% to 38% in most countries in Europe, with Portugal, Poland, Romania and Cyprus having the highest rates [11]. More recent data show that the rate of intervention during birth is continuing to increase: in 2020/2021, 33.5% of all births in NHS hospitals in England were caesarean sections [12]. Increasing rates of caesarean sections can also be seen over the years in other European countries [13]. In North America, the caesarean section rate is 31.6%; in Latin America and the Caribbean it is 42.8%; on the continent of Africa, it is 9.2%; however, northern Africa has a caesarean section rate of 32.0%, whilst it is 5.0% in Sub-Saharan Africa [14]. In Brazil, caesarean section rates are higher than in many other countries (55.9%), with particularly high rates being observed in the private healthcare sector (72%) [15,16,17].

Despite the significant amount of literature surrounding the impact of mode of birth on postpartum sexual functioning in the short-and longer-term, there is no consensus on the association between mode of birth and sexual functioning. Several cohort studies have shown that women who have a spontaneous vaginal birth (with an intact perineum) are more likely to resume vaginal sex by six to eight weeks postpartum compared to women who have an episiotomy, an assisted vaginal birth or a caesarean section [18,19]. Another cohort study showed that, by two months postpartum, 43% of women who had a caesarean section had at least one problem whilst attempting intercourse, and that assisted vaginal birth was significantly associated with perineal pain persisting after two months postpartum when compared to spontaneous vaginal delivery [20]. However, a cohort study did not show an association between mode of birth and sexual function in the short term [21]. A cohort study showed that 35% of women suffered loss of sexual desire up to 8–9 months after birth [22]. At 16 years postpartum, a cohort study showed that, on average, women who had caesarean section had an increased frequency of sexual problems when compared to other modes of birth; however, no difference was seen in the prevalence of sexual problems when comparing spontaneous and assisted vaginal birth [23]. However, it should be noted that other factors, such as menopause, mental health and physical health, could impact postpartum sexual function in the longer-term [24,25,26,27]. Sexual (dys)function prior to pregnancy, depressive symptoms, breastfeeding status, parity and perineal tears are other potential confounders [3,20,28,29,30,31,32,33].

Systematic reviews on the association between mode of birth and postpartum sexual function have been undertaken in 2004, 2010, 2020 and 2022 [3,34,35,36]. Results from these systematic reviews varied: a review showed that caesarean section was protective of sexual function compared to vaginal birth [34], another showed a strong association between assisted vaginal birth and worse sexual function when compared to other modes of birth [35], whilst others showed no relationship between mode of birth and postpartum sexual function [3,36]. However, these reviews had several methodological shortcomings: the studies included in the reviews by Sayasneh et al. (2010) and Hicks et al. (2004) had a high heterogeneity, and they were included regardless of whether they assessed sexual functioning with a validated measure [34,35]; the review by Gutzeit et al. (2020) lacked a clear methodology, reporting no inclusion and exclusion criteria [3]; the review by Nikolaidou et al. (2022) included all available studies on the association between type of birth and female sexual function up until 12 months following the birth using established, validated tools [36], but they did not consider episiotomy in their review, which is a common birth intervention that can have a major impact on sexual functioning [11,19].

Sexual dysfunction that continues beyond six months postpartum is more likely to have an impact on maternal wellbeing: women with impaired sexual function are more likely to score worse in all quality of life domains; and women at six months postpartum with worse sexual functioning scores are more likely to have worse scores for general health and physical functioning [2,4]. Therefore, the aim of this integrative review was to examine the impact of different modes of birth, as well as episiotomy, on sexual functioning, beyond six months postpartum in the medium-term (≥6 months and <12 months postpartum) and longer-term (≥12 months postpartum).

## 2. Materials and Methods

### 2.1. Search Strategy

Relevant studies were identified using PubMed, Embase and CINAHL databases, and searches were limited to studies published between January 2000 and February 2023. Keywords in the search included: mode of birth (e.g., vaginal birth, caesarean section), episiotomy, postpartum, and sexual function. A detailed search strategy is presented in the Appendix B. The reference lists of the studies that were selected for inclusion were also manually searched for additional studies. This systematic review was registered in PROSPERO (CRD42020212746).

### 2.2. Inclusion and Exclusion Criteria

Both quantitative and mixed method studies were eligible for inclusion in this review, including cohort studies, cross-sectional studies, case–control studies and randomised controlled trials. Systematic reviews, unpublished results or studies, protocols, studies not written in English or written prior to 2000 were excluded. In order to be included, studies must have compared at least two different modes of birth. To facilitate clinical usability of this review, as well as provide a useful framework for this review and other studies in the future, specific modes of birth were the main focus of the review. Therefore, studies that compared data on spontaneous vaginal birth, vaginal birth with episiotomy, assisted vaginal birth, planned (elective) caesarean section and emergency caesarean section were included. The secondary focus of this review were comparisons that incorporated multiple modes of birth in each category, as some studies combine multiple modes of birth into one group. Therefore, studies that compared non-specific mode of birth groups such as vaginal birth (without differentiation between spontaneous vaginal birth, vaginal birth with episiotomy or assisted vaginal birth), vaginal birth with breech presentation, and caesarean section (without differentiation between planned and emergency caesarean section) were included. Only episiotomy was considered, rather than other perineal trauma, as it is an intervention performed (or not) by a health care professional during birth, whereas other perineal trauma happens as a result of birth. In order to ensure comparability of the studies, only studies that used a validated sexual function measure were included.

### 2.3. Valid Measures of Sexual Function

All of the measures of sexual function used in the studies included in this review are self-reported measures. The measures contain questions/statements, to which women report on their sexual function, by rating their sexual function using a numeric/Likert scale. A measure was considered to be a valid measure for sexual function if it has undergone psychometric evaluation and its psychometric properties were found to be acceptable.

The Female Sexual Function Index (FSFI) consists of 19 items embedded in six domains (desire, arousal, lubrication, orgasm, satisfaction and pain). The theoretical range is 2.0 to 36.0, with higher total FSFI-scores indicating better sexual functioning [37]. A cut off of ≤26.55 is commonly used to report a dichotomous FSFI-score indicating sexual dysfunction [38]. The FSFI has been evaluated in terms of psychometric quality, which was found to be good in women with varying parity [37,39].

The Golombok Rust Inventory of Sexual Satisfaction (GRISS) for women consists of 28 items embedded in seven subscale domains (non-communication, infrequency, dissatisfaction, avoidance, non-sensuality, vaginismus, anorgasmia). The raw score is converted into a transformed score ranging between one and nine, which can be used to produce a diagnostic profile. A transformed score greater than or equal to 5 indicates sexual dysfunction [40,41,42]. The GRISS has been evaluated in terms of psychometric quality, which was found to be good [40].

The Sexual Health Outcomes in Women Questionnaire (SHOW-Q) has 12 items embedded in four subscales (satisfaction, orgasm, desire, pelvic problem interference). It was designed to be used on a diverse range of women, including women in same sex relationships, women who are sexually active without a partner, and sexually inactive women. All items are scored on a scale from zero to 100. For three of the subscales (satisfaction, orgasm, desire), higher scores represent better sexual function. For the final subscale (pelvic problem interference), higher scores indicate worse sexual function. After reversal of the pelvic problem interference scores, all items are totalled for an overall score with a theoretical range of zero to 400 [43]. Lower total scores indicate worse sexual functioning [43,44]. The SHOW-Q has been evaluated in terms of psychometric quality, which was found to acceptable in women with varying parity [44].

The Sexual Activity Questionnaire (SAQ) has 21 items embedded in three scales (pleasure, discomfort and habit). Each question has weighted loadings to create factor scores, with high scores in the pleasure domain representing high pleasure, and low scores in the discomfort domain representing low levels of discomfort. The habit domain is a single question and the value given as the answer is the score for this domain, with more habitual regular sexual activity resulting in higher scores. Each question is rated using a four-point Likert scale from 0 to 3, with the total score ranging from 0 to 30 [45,46]. The SAQ has been evaluated in terms of psychometric quality, which was found to be good [47,48].

The Sexual Function Questionnaire (SFQ28) has 28 items embedded in eight domains (desire, arousal (sensation), arousal (lubrication), arousal (cognitive), orgasm, pain, enjoyment, partner (questions regarding women’s opinion of their sexual life with their partner)). Lower scores indicate increased probability of sexual dysfunction. Overall total score has a theoretical range of 14 to 170. Each domain is scored separately, with different ranges of scores indicating sexual dysfunction. For all domains, lower scores indicate worse sexual functioning [49,50]. The SFQ28 has been evaluated in terms of psychometric quality, which was found to be good [51].

The Pelvic Organ Prolapse/Urinary Incontinence Sexual Questionnaire (PISQ-12) is a 12-item questionnaire that measures three domains: behavioural–emotive, physical and partner/related (questions regarding women’s opinion of their sexual life with their partner). Each item is graded on a five-point Likert scale from always (zero) to never (zero). The behavioural–emotive items are reversely scored. Higher scores indicate better sexual functioning, with a maximum total score of 48 [52,53]. The PISQ-12 has been evaluated in terms of psychometric quality, which was found to be good in women with varying parity [52,54].

### 2.4. Classification of the Medium- and Longer-Term Studies

Results from the sexual function measures were categorised into medium- and longer-term: studies that collected data at beyond or equal to six months postpartum and at less than 12 months postpartum were classified as medium-term, whereas those that collected data at beyond or equal to 12 months postpartum were classified as longer-term.

### 2.5. Selection of the Studies

Studies that were identified as potentially eligible for inclusion were exported to Rayyan, a systematic review reference management tool. Duplicates were removed. When deciding which studies to include, two researchers (AMF, NG) independently screened the titles and abstracts. From there, full texts were evaluated for their eligibility by the two researchers, based on inclusion and exclusion criteria. Any inconsistency between researchers on whether studies should be included or not were discussed further in order to come to an agreement. If agreement could not be reached, a third author (LP) was involved in the discussion. Information on study design, location of study, descriptive characteristics of participants, types of modes of birth, the measure of sexual function, and the results of those studies related to mode of birth and sexual function was then extracted.

### 2.6. Quality Assessment

Quality of the included studies was assessed by two researchers (AMF, NG) independently using the Newcastle Ottawa Scale (NOS) [55]. The NOS tool was “developed to assess the quality of non-randomised studies” for use in systematic reviews and meta-analyses. The NOS has three domains: selection, comparability and exposure. The selection domain covers representativeness of the studied population in the community, selection of the non-exposed cohort, ascertainment of exposure, and demonstration that the outcome of interest was not present at start of the study [55]. The comparability domain assesses control for confounding variables. The outcome domain covers assessment of the outcome, length of the follow up and adequacy of follow up. Each domain contains several questions with multiple choice answer options. A maximum of one star can be awarded for each question [55]. We adapted the NOS, as self-reported measures of sexual function were considered the gold standard for assessing sexual measures by the authors, since sexual function can only be reported by the person experiencing it. The NOS was adapted by awarding a star for the assessment of outcome question, in the outcome domain.

A scoring system developed by Sharmin et al. (2017) was used to group the included studies by quality score: a study was rated good quality if it scored three or four stars in selection, one or two stars in comparability, and two or three stars in outcomes. A study was rated fair quality if it scored two stars in selection, one or two stars in comparability and two or three stars in outcome. A study was rated poor quality if it scored zero or one star in selection, or zero stars in comparability, or zero or one star in outcomes [56,57]. Tabular explanations of the domains and the scoring system are presented in Appendix A, respectively.

### 2.7. Data Extraction and Analysis

Data from the included studies was extracted and reported using the Population, Exposure, Comparator and Outcomes (PECO) framework [58]. Extracted information included the author(s), year of publication, country, population (number of participants, ethnicity, age, parity), exposure/comparator (mode of birth), outcomes (measure used to assess the level of sexual function, interval from birth to sexual function measurement) and the main findings. The main findings show the impact between modes of birth and sexual function with 1) associations (crude or adjusted): this includes odds ratios with 95% confidence intervals, or relative risk with 95% confidence intervals 2) differences between modes of birth and accompanying mean (standard deviation) or median (interquartile range) sexual function scores.

## 3. Results

### 3.1. Search

In total, 6509 studies were identified in database searches. After removing the duplicates, screening of the titles and abstracts against the inclusion and exclusion criteria was carried out, as well as checking references and similar articles, 112 studies remained. These studies were further evaluated based on the full texts, of which 31 were included in the review. A detailed overview of the search, screening and selection process is presented in a PRISMA flowchart (Figure 1). A list of the studies that were excluded after reading the full text, and the reasons for their exclusion, is presented in Appendix A.

### 3.2. Study Characteristics

Of the 31 included studies, 9 were cross-sectional studies (4966 women; including at least 1591 primiparous women), 3 were case–control studies (368 women; including at least 357 primiparous women), 5 were retrospective cohort studies (1872 women; including at least 474 primiparous women) and 14 were prospective cohort studies (4217 women; including at least 3338 primiparous women). The studies were published between 2005 and 2022, originating from 17 countries including Turkey [59,60,61,62,63,64], the United States of America [65,66,67,68], Iran [69,70,71,72], Australia [73,74], Austria [75], Egypt [76], Poland [77], Germany [78], Japan [79], Italy [80,81], Switzerland [82], Hungary [83], China [84], Taiwan [85], Sweden [86], Israel [87] and Portugal [88]. One study did not disclose the country of origin of the research [89]. Sample size ranged from 53 to 2765. Detailed information about the included studies is presented in Table 1 (studies are sorted by study design in the order: cross-sectional, case–control, retrospective cohort, prospective cohort).

In the included studies, six different measures were used to assess sexual functioning. The FSFI was used in 25 studies [59,60,62,63,64,65,66,67,69,70,71,72,74,75,76,77,80,81,82,83,84,85,87,88,89]. A combination of the FSFI and SAQ was used in one study [78]. The GRISS was used in 2 studies [61,73]. The SHOW-Q was used in one study [68]. The SFQ28 was used in one study [79]. The PISQ-12 was used in one study [86]. Data in the included studies were collected by administering the valid measures of sexual function in the following ways: four studies used the postal system [73,74,79,82], nine studies used face to face structured interviews, or self-reporting in person at the clinics [59,60,61,62,63,64,71,72,84], two studies used telephone interviews to complete the questionnaire [65,87], four studies used online forums or web/based questionnaires [77,83,86,88], one study used a combination of telephone and in person questionnaires [68], and eleven studies did not specify how they administered the measures to the participants [66,67,69,70,75,76,78,80,81,85,89].

Of the 31 studies, 18 measured sexual function in the medium-term at ≥six months and <12 months postpartum (4945 women; including at least 3409 primiparous women) [62,63,64,67,68,69,71,74,77,78,79,80,81,83,85,87,88,89], 20 measured sexual function in the longer-term at ≥12 months postpartum (8643 women; including at least 3887 primiparous women) [59,60,61,62,65,66,70,71,73,74,75,76,78,82,83,84,85,86,88,89] and eight measured sexual function in both the medium-term and longer-term (2378 women; including at least 1749 primiparous women) [62,71,74,78,83,88,89,90]. One study measured sexual function between six and 24 months postpartum and did not provide an average time when sexual function was measured (213 women; all 213 women were primiparous) [72].

When comparing studies in terms of parity, 16 studies included only primiparous women (3901 primiparous women) [60,61,62,64,65,67,69,70,71,72,74,75,80,84,86,89], 14 studies included both primiparous and multiparous women (7222 women; including at least 1859 primiparous women) [59,63,66,68,73,77,78,79,81,82,83,85,87,88] and one study did not disclose the parity of the women included (300 women) [76].

There were 17 different modes of birth comparisons; the most common comparison was between vaginal birth (without differentiation between spontaneous vaginal birth, vaginal birth with episiotomy and assisted vaginal birth) and caesarean section (without differentiation between planned and emergency caesarean section) (Table 1). None of the included studies compared modes of birth in cases of breech presentation.

### 3.3. Quality Assessment

Of the included studies, 21 were of a good quality, nine studies were rated as having a fair quality and one study was rated as being of a poor quality (Table 2).

### 3.4. Main Focus: Specific Mode of Birth Groups (SVB, VBE, AVB, pCS, eCS)

#### 3.4.1. Medium-Term Outcomes (≥Six Months and <12 Months Postpartum)

Three studies showed three statistically significant associations between mode of birth and sexual function in the medium-term (Table 3) [64,77,80].

One study showed a statistically significant adjusted association between mode of birth and sexual function in the medium-term. Planned caesarean section was associated with better total sexual functioning scores and better scores for arousal, lubrication and orgasm when compared to assisted vaginal birth [80].

Two studies showed two statistically significant crude associations between mode of birth and sexual function in the medium-term. Spontaneous vaginal birth was associated with better total sexual functioning scores when compared to assisted vaginal birth [77]. Planned caesarean section was associated with better total sexual functioning scores and better scores for arousal, lubrication, orgasm, satisfaction and pain when compared to vaginal birth with episiotomy [64] (Table 3).

Six studies that showed no statistically significant associations between mode of birth and sexual function in the medium-term [62,69,78,83,87,88] (Table 3).

#### 3.4.2. Longer-Term Outcomes (≥12 months Postpartum)

Six studies showed 25 statistically significant associations between mode of birth and sexual function in the longer-term [60,65,76,82,86,88] (Table 4).

Two studies showed two statistically significant adjusted associations between mode of birth and sexual function in the longer-term. Assisted vaginal birth was associated with increased odds of pain at 12 months postpartum, in comparison with spontaneous vaginal birth [86]. Planned caesarean section was associated with an increased risk of pain and loss of desire at six years postpartum when compared to spontaneous vaginal birth [82].

Four studies showed 23 statistically significant crude associations between mode of birth and sexual function in the longer-term. Spontaneous vaginal birth was associated with better scores pain when compared to assisted vaginal birth [88]. Vaginal birth with episiotomy was associated with worse overall sexual function and worse scores for arousal and orgasm when compared to planned caesarean section [60]. Assisted vaginal birth was associated with worse scores for orgasm and satisfaction when compared to planned caesarean section [65]. One study showed mixed results for associations between modes of birth and sexual function [76] (Table 4).

Four studies showed no statistically significant associations between mode of birth and sexual function in the longer-term [62,75,78,83] (Table 4).

### 3.5. Secondary Focus: Non-Specific Mode of Birth Groups (Incorporated Multiple Mode of Birth Groups into One Category)

#### 3.5.1. Medium-Term Outcomes (≥Six Months and <12 Months Postpartum)

Seven studies showed 12 statistically significant associations between mode of birth and sexual function in the medium-term (Table 5) [67,71,77,79,80,81,89].

Four studies showed five statistically significant adjusted associations between mode of birth and sexual function in the medium-term. Vaginal birth with episiotomy was associated with worse total sexual functioning scores and worse sexual functioning scores for desire, compared to caesarean section (without differentiation between planned and emergency caesarean section) [71]. Assisted vaginal birth was associated with worse sexual functioning scores for orgasm when compared to vaginal birth (without differentiation between spontaneous vaginal birth and vaginal birth with episiotomy) [80]. Caesarean section (without differentiation between planned and emergency caesarean section) was associated with better sexual functioning scores for arousal, desire and lubrication when compared to assisted vaginal birth [79]. Vaginal birth (without differentiation between spontaneous vaginal birth, vaginal birth with episiotomy and assisted vaginal birth) was associated with better total sexual functioning scores when compared to caesarean section (without differentiation between planned and emergency caesarean section) [67], but was associated with worse sexual functioning scores for arousal, desire and lubrication when compared to caesarean section (without differentiation between planned and emergency caesarean section) [79].

Four studies showed seven statistically significant crude associations between mode of birth and sexual function in the medium-term. Spontaneous vaginal birth was associated with better sexual functioning scores for arousal, compared to vaginal birth (without differentiation between vaginal episiotomy and assisted vaginal birth) and caesarean section (without differentiation between planned and emergency caesarean section) [77]. Caesarean section (without differentiation between planned and emergency caesarean section) was associated with better total sexual functioning scores and better scores for lubrication and orgasm yet was associated with worse scores for pain when compared to vaginal birth with episiotomy [89]. Assisted vaginal birth was associated with better scores for sexual satisfaction with partner, compared to vaginal birth (without differentiation between spontaneous vaginal birth and vaginal birth with episiotomy [79]. Caesarean section (without differentiation between planned and emergency caesarean section) was associated with better total sexual functioning scores and better scores for pain and arousal, compared to vaginal birth (without differentiation between spontaneous vaginal birth, vaginal birth with episiotomy and assisted vaginal birth) [77,81] (Table 5).

Six studies showed no statistically significant associations between mode of birth and sexual function in the medium-term (Table 5) [63,68,74,78,85,88].

#### 3.5.2. Longer-Term Outcomes (≥12 Months Postpartum)

Four studies showed 6 statistically significant associations between mode of birth and sexual function in the longer-term (Table 6) [66,70,71,73].

Two studies showed four statistically significant adjusted associations between mode of birth and sexual function in the longer-term. Caesarean section (without differentiation between planned and emergency caesarean section) was associated with increased satisfaction with vaginal tone when compared to spontaneous vaginal birth and assisted vaginal birth [73]. Assisted vaginal birth was associated with an increased risk of loss of desire at ≥10 years postpartum, when compared to vaginal birth (without differentiation between spontaneous vaginal birth and vaginal birth with episiotomy), and caesarean section (without differentiation between planned and emergency caesarean section) [66].

Two studies showed two statistically significant crude associations between mode of birth and sexual function in the longer-term. Caesarean section (without differentiation between pCS and eCS) was associated with better scores for desire when compared to vaginal birth with episiotomy [71], and better scores for arousal when compared to vaginal birth (without differentiation between spontaneous vaginal birth, vaginal birth with episiotomy and assisted vaginal birth) [70].

Seven studies showed no statistically significant associations between mode of birth and sexual function in the longer-term (Table 6) [59,61,74,78,84,85,86,89].

### 3.6. Overall

Various factors were considered important when comparing the studies included in this review: study design, sample size, parity, adjustment for confounders, longitudinal follow up and study quality. Comparative analyses were performed to evaluate whether associations varied between the studies that differed in these factors.

When comparing the study design, a similar proportion of the studies were cohort or longitudinal studies compared to cross-sectional studies, and both categories showed mixed results without any consistent findings. There were more studies with a sample size of less than 500 participants [59,60,61,62,63,64,65,67,68,69,70,71,72,74,75,76,77,79,80,81,83,88] than those with a larger sample size (≥500) [66,73,78,82], yet no difference was seen in the results. There were no statistically significant associations between vaginal birth with episiotomy and sexual function, compared to other modes of birth, in studies that included both primiparous and multiparous women [59,63,66,68,73,77,78,79,81,82,83,87,88], whereas there were significant associations in the studies that included only primiparous women [60,61,62,64,65,67,69,70,71,72,74,75,80,84,85,86,89].

Based on the existing literature on sexual function, the confounders considered of great importance include: sexual (dys)function prior to pregnancy, (risk of) depression, breastfeeding status, parity and perineal tears [3,20,28,29,30,31,32,33]. Of the 15 studies that adjusted for any confounders in the analysis, nine adjusted for at least one of the most important confounders [68,73,74,77,80,82,83,85,86]. The other six studies adjusted for demographic and general health characteristics only [66,67,71,72,75,79]. No difference was seen in the results of the studies that adjusted for most important confounders and those that did not.

Of the longitudinal studies that were included, some showed no differences in associations between mode of birth and sexual functioning across various time points [62,74,78,83,85]. Some studies showed improved sexual function scores over time [71,77,89]. One longitudinal study showed worse sexual function scores for assisted vaginal births compared to spontaneous vaginal births for the domain pain at 12 months postpartum. This association was, however, not present at six months postpartum [88].

When comparing the results based on the quality of studies, no differences were seen in associations between studies that were of poor, fair and good quality.

## 4. Discussion

The main focus of our review was to examine the impact of specific mode of birth on medium- and longer-term sexual functioning. Our review shows that there are significant associations between mode of birth and women’s sexual function in both the medium- and longer-term. Adjusted associations showed that: assisted vaginal birth was associated with worse sexual function (overall sexual function, and arousal and lubrication domains) when compared to planned caesarean section, in the medium-term (≥six months and <12 months postpartum); in the longer-term (≥12 months postpartum), spontaneous vaginal birth was associated with better sexual function when compared to assisted vaginal birth (pain domain) and planned caesarean section (desire and pain domains). In both the medium- and longer-term, assisted vaginal birth was consistently associated with worse sexual function, compared to other modes of birth.

In regard to the secondary focus of this review (multiple modes of birth combined into one groups), no consistent pattern was shown by adjusted associations to reflect that one mode of birth group was associated with better/worse sexual function compared to another, in the medium- or longer-term.

The results from both the main and secondary focus of this review suggest that sexual dysfunction does not necessarily resolve in the first year postpartum, and that assisted vaginal birth can have lasting effects on sexual function past 12 months postpartum.

Many studies included in this review show that mode of birth is not associated with postpartum sexual functioning in the medium- and longer-term. This supports the findings of Nikolaidou in their systematic review, on the effect of mode of birth on sexual function in the short- and medium-term (up until 12 months postpartum) [36].

The associations shown in this review between assisted vaginal birth, as well as vaginal birth with episiotomy, and worse sexual function can be linked to mechanisms of injury surrounding these operative interventions: assisted vaginal birth is associated with a higher incidence of episiotomy and severe perineal trauma, as well as pelvic floor trauma and nerve damage which can affect sexual function [91,92,93,94,95]. Worse sexual function after planned caesarean section compared to spontaneous vaginal birth could be contributed to by the increased probability of wound problems/scar pain due to the abdominal incision that is involved in caesarean sections as wound problems (both relating to caesarean abdominal wounds and perineal trauma in spontaneous vaginal births) are associated with impaired general sexual health [96]. The protective effect on sexual function that was seen with planned caesarean section could be because women who undergo caesarean section have less pelvic floor trauma compared to those who have had a vaginal birth [95,97,98,99]. However, it must be noted that caesarean section is a major operation that can have serious implications for maternal health and should not be considered solely to preserve sexual function [100].

The included studies varied by study design, sample size, adjustment for confounders, and study quality, however the associations seem not to be affected by these differences.

The longitudinal improvement in sexual function could be due to the natural course of recovery after birth [101]. On the other hand, the worsening of sexual function over time could indicate that, if left untreated, postpartum sexual pain could get worse for those who had an assisted vaginal birth. However, worsening of postpartum sexual function over time could also be related to other postpartum risk factors such as breastfeeding, fatigue or stress [102].

### 4.1. Strengths and Limitations

The main strength of this integrative review is that only studies using validated measures to assess sexual functioning were included, allowing direct comparison between the studies, as well as increasing the ability to draw clinical implications from the results. Multiple mode of birth and intervention combinations were investigated. Additionally, we focused on the effects beyond six months postpartum, and did not limit the period of interest to a certain time after birth, meaning studies were included that examined associations up to and greater than 10 years postpartum. The majority of studies included in this review were of good quality.

Despite the strengths of this review, some limitations should be noted. When calculating associations between mode of birth and sexual function, some of the included studies did not adjust for important confounding factors in the analysis. Sexual function can be influenced by many factors, including postpartum stress and fatigue, managing relationships with partner (if applicable), life stressors (such as relationship with family, friends and work), breastfeeding, experiences during previous births, history of sexual abuse and degree of sexual dysfunction before pregnancy and birth [3,20,28,29,30,31,32,33]. Without accounting for these factors, any associations, or lack of, may be misrepresented.

Despite the evidence showing the difference between male and female sexual function, some measures of sexual function are based on dated evidence which suggests that women’s sexual function is equivalent to men’s sexual function [103], even though this evidence has been found to ignore “major components of women’s sexual satisfaction” [104], and is a more appropriate reflection of male sexuality [8]. In addition, the measures used by the studies included in this review have not yet been validated in postpartum women.

An additional limitation that should be noted is the lack of consensus on reporting standards in the papers: reporting for mode of births, as well as adjustment for confounding, was not consistent across the papers included in this review.

Two studies examined sexual function at and beyond six years postpartum and found significant adjusted associations with mode of birth [66,73]. Although it is important to examine longer-term postpartum sexual functioning, it should be noted that such a long time gap between birth and measurement of postpartum sexual function means that other factors, besides mode of birth and those adjusted for, may affect sexual function (e.g., menopause, mental health, physical health relationship satisfaction, relationship with child) [3,20,24,25,26,27,28,29,30,31,32,33].

Many of the included studies did not adjust for important confounders, with only nine of 31 studies adjusting for at least one of the most important confounders. Therefore, this needs to be considered as a limitation and taken into account when interpreting any associations reported in this review.

### 4.2. Recommendations

Future studies should take into account pre-existing sexual dysfunction, quality of the relationship between the parents before and after birth, experienced birth trauma and the model of care experienced, in order to elucidate whether these factors can explain the associations found between mode of birth and sexual function.

Another avenue of research that should be explored is how the health care a woman receives, if any, during the postpartum period affects sexual function. Given how important sexual functioning can be for a woman [1,2,3,4], future research should aim to identify types of care that can help avoid a decline in postpartum sexual function.

Additionally, studying women who have more complex births and are underrepresented in the current literature should be made a priority: for example, many studies excluded women who did not have a singleton birth despite the fact that the proportion of twin births out of total births (per 1000 births) is 14.4 in Europe [105]. In a similar vein, some studies excluded births of babies that were not in cephalic presentation, despite the fact that approximately three to five per cent of women give birth to a child in a breech presentation [106,107].

There is some evidence that incidence of perineal trauma after birth varies between different ethnic groups; a systematic review showed that the majority of studies conducted in Western countries describe women of Asian ethnicity as having increased risk of severe perineal trauma during birth, whereas Asian women living in Asia were not at increased risk. The reasons for this were unknown but could be attributed to differing care that women from different ethnic groups receive, differing birthing techniques, ethnic anatomical differences or cultural differences, and changes in diet when moving to Western countries [108,109]. Therefore, the diversity of the study population, within the geographical area that the study is conducted in, should be disclosed and associations between different ethnicities should be explored in order to attempt to explain why some ethnicities can be at higher risk during labour and birth, as well as during the postpartum period.

It would be valuable to contextualise sexual functioning scores by asking women more in depth, follow-up questions about their postpartum sexual function. This would help to understand why associations exist in the quantitative data, and identify how postpartum care, focused on sexual function, could help women. It is of vital importance that measures are validated in postpartum women. Additionally, it is important that sexual function measures and have undergone cross-cultural validation, for both linguistics and culture, when they are used to measure sexual function in women whose cultural backgrounds differ from those that the measures were originally validated in.

## 5. Conclusions

This integrative review summarises the existing evidence on the impact of different modes of birth on sexual functioning, in the medium- (≥six months and <12 months postpartum) and longer-term (≥12 months postpartum), published between 2005 and 2023. Assisted vaginal birth has been found to be associated with worse sexual function when compared to spontaneous vaginal birth and caesarean section in both the medium- and longer-term. However, it should also be noted that some studies showed that spontaneous vaginal birth and caesarean section were associated with worse sexual function in comparison to other modes of birth. No mode of birth can be recommended to preserve sexual function in the medium- or longer-term. Therefore, it is important that postpartum sexual functioning is discussed with mothers for all modes of births, in order to encourage women to seek help and support from healthcare professionals should their sexual function be impaired after birth.

## Figures and Tables

**Figure 1 ijerph-20-05252-f001:**
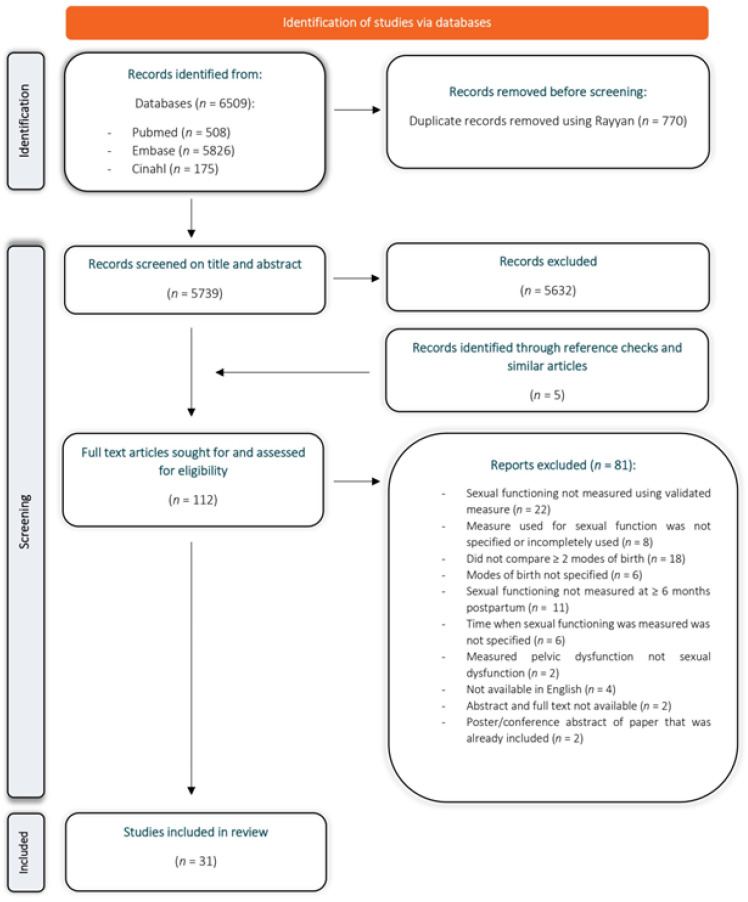
PRISMA flowchart illustrating the systematic database search for articles that met the inclusion criteria.

**Table 1 ijerph-20-05252-t001:** Characteristics of included studies.

					Population		Exposure/Comparison		Outcome
Author, Year	Population Characteristics	Study Design	Data Collection	Time of Data Collection	Analysed Participants	Mode of Birth	Definition Sexual Health/ Function	Statistical Analysis	Results
Barbara et al. 2016 [80]	Country: Italy Ethnicity: Caucasian Age: Mean = 34.4 (SD ± 4.9) Parity: 100% primiparous	Cross sectional	How: Questionnaire Inclusion rate: 53.3% (269/505)	6 months postpartum	Number: *n* = 269 Recruitment: During first 3 days after birth before discharge from Department of Women’s and Children’s Health, Fondazione IRCCS Ca’ Granda, Ospedale Maggiore Policlinico, Milan, Italy in 2013.	SVB/ VBE: 49.1% (*n* = 132) AVB: 16.7% (*n* = 45) pCS: 34.2% (*n* = 92)	FSFI total score FSFI domain scores FSFI dichotomous score: sexual dysfunction was defined with a score ≤26.5	Analysis: One-way and multivariate ANOVA and Chi-square test Multiple hierarchical regressions: To control for confounders (demographics, postpartum depression and breastfeeding status)	At 6 months after birth, sexual dysfunction was highest in AVB group (51.1%) compared to SVB/VBE (42.2%) and pCS (33.7%) groups. After multiple hierarchical regression: At 6 months after birth, there was association between AVB and worse total FSFI score, compared to pCS. (β = −2.566, SE = 0.995, 95% CI= −4.525, −0.607, *p* = 0.01). At 6 months after birth, there was association between AVB and worse arousal FSFI domain score, compared to pCS. (β = −0.471, SE = 0.192, 95% CI = −0.849, −0.093, *p* = 0.015) At 6 months after birth, there was association between AVB and worse lubrication FSFI domain score, compared to pCS. (β = −0.451, SE = 0.226, 95% CI = −0.897, −0.005, *p* = 0.05) At 6 months after birth, there was association between AVB and worse orgasm FSFI domain score, compared to pCS. (β = −0.627, SE = 0.253, 95% CI = −4.525, −0.607 *p* = 0.01) At 6 months after birth, there was association between SVB/VBE and better orgasm FSFI domain score, compared to AVB. (β = 0.605, SE = 0.233, 95% CI = 0.145, 1.064, *p* = 0.01). No significant differences were found between SVB/VBE and pCS groups for total FSFI score or FSFI domain specific scores.
Baud et al., 2020 [82]	Country: Switzerland Ethnicity: Not reported Age (at time of questionnaire, approximately 6 years postpartum): SVB: Mean = 36.6 (SD ± 5.3) pCS: Mean = 37.3 (SD ± 5.4) Parity: SVB: 83.7% primiparous; 16.3% multiparous pCS: 81.5% primiparous; 18.5% multiparous	Cross sectional	How: Postal questionnaires Inclusion rate: 49% (604/1231)	6 years postpartum	Number: *n* = 517 Recruitment: Women who gave birth in period 1996–2011 whose data was in the obstetrical database at the Maternity Hospital of the Centre Hospitalier Universitaire Vaudois.	SVB: 59.8% (*n* = 309) pCS: 40.2% (*n* = 208)	FSFI -total score FSFI dichotomous score: severe sexual dysfunction was defined with a score ≤25	Analysis: Generalised linear models using Poisson regression with robust variance estimates. Linear modelling: Poisson regression with robust variance estimates. Relative risks were adjusted for all sociodemographic and physical variables. Following confounders were considered: obstetrical exposures, age, ethnicity, parity, weight, smoking, level of education, marital status, religion, type of health insurance.	The occurrence of severe sexual dysfunction was statistically significantly higher after pCS than SVB (36.2% versus 26.4%, respectively). Median FSFI score for all items investigating pain during or following sexual intercourse were significantly worse after pCS (median score = 4.78 (IQR not reported) than SVB (median score = 5.14 (IQR not reported). *P* = 0.002 After adjustment for all sociodemographic and physical variables: Item 2 of the FSFI (low level of sexual desire) showed a significantly increased risk of presenting symptoms after pCS compared to SVB (aRR = 1.54, 95% CI = 0.96–2.50, *p* = 0.041). Item 17 of the FSFI (pain during vaginal penetration about half the time or more) showed a significantly increased risk of presenting symptoms after pCS compared to SVB (aRR = 2.04, 95% CI = 1.10–3.85, *p* = 0.001). Item 18 of the FSFI (pain following vaginal penetration more than half the time) showed a significantly increased risk of presenting symptoms after pCS compared to SVB (aRR = 1.79, 95% CI = 1.04–3.03, *p* = 0.027). Item 19 of the FSFI (high level of pain following vaginal penetration) showed a significantly increased risk of presenting symptoms after pCS compared to SVB (aRR = 2.50, 95% CI = 1.19–5.26, *p* = 0.002).
Dabiri et al. 2014 [69]	Country: Iran Ethnicity: Not reported Age: VBE: Mean = 27.87 (SD ± 5.64) pCS: Mean = 28.2 (SD ± 3.64) Parity: 100% primiparous	Cross sectional	How: Self-completed questionnaire Inclusion rate: Not reported	6 months postpartum	Number: *n* = 150 Recruitment: Primiparous women who had brought their babies for vaccination/attended the family planning clinics at health and teaching clinics in Bandar Abbas from September 2010–April 2011.	VBE: 54% (*n* = 81) pCS: 46% (*n* = 69)	FSFI total score FSFI domain scores.	Analysis: Independent samples of *t* test for comparison between groups and paired *t* test for comparison within groups.	No significant difference in total FSFI scores and domain specific FSFI scores between pCS and VBE groups.
Dean et al. 2008 [73]	Country: Australia Ethnicity: Not reported Age: Mean = 35 (SD not reported) Parity: Mean parity = 2.4	Cross sectional	How: Postal questionnaire Inclusion rate: 35% (2765/7872)	6 years postpartum	Number: *n* = 2765 Recruitment: All women who delivered between January 1994 and March 1995 in 3 maternity units (Scotland, England, New Zealand).	SVB: 51.3% (*n* = 1419) AVB: 30.8% (*n* = 852) CS: 9.7% (*n* = 9.7%)	Non-transformed GRISS domain scores (modified)	Analysis: three-way ANOVA ANOVA: Adjustment for parity, current pelvic floor muscle exercises, urinary and faecal incontinence.	Women who had CS had a significantly better mean GRISS score (mean = 3.46, SD ± 1.02) when rating their vaginal tone for their own satisfaction, when compared with SVB/VBE (mean = 3.21, SD ± 0.99) and AVB groups (mean = 3.17, SD ± 1.09). *p* ≤ 0.001 Women who had CS had a significantly better mean GRISS score (mean = 3.66, SD ± 0.98) when rating their vaginal tone for their partner’s satisfaction, when compared with SVB/VBE (mean = 3.45, SD ± 0.97) and AVB groups (mean = 3.41, SD ± 1.02). *p* ≤ 0.01. After adjustment: Results remained significant after adjustment.
Ghorat et al. 2017 [70]	Country: Iran Ethnicity: Not reported Age: Mean = 31.81 (SD ± 6.31) Parity: 100% primiparous	Cross sectional	How: Questionnaire Inclusion rate: Not reported	2 years postpartum	Number: *n* = 177 Recruitment Random sampling from every primiparous woman who had delivered 2 years before 2014, in Sabzevar Hospitals.	SVB/ VBE: 69.5% CS: 30.5%	FSFI total score FSFI domain scores FSFI dichotomous cut off score for sexual dysfunction: 28 FSFI domain dichotomous cut off scores for sexual dysfunction: Sexual pain = 3.8 Sexual desire = 3.3 Sexual arousal = 3.4 Lubrication = 3.7 Orgasm = 3.4 Sexual satisfaction = 3.8	Analysis: Chi square, student t-test and Fischer exact test.	Mean FSFI score of arousal domain was significantly worse in SVB/VBE group (mean = 3.39, SD ± 1.11) compared to CS group (mean = 3.76, SD ± 1.1), *p* ≤ 0.05.
Hosseini et al. 2012 [72]	Country: Iran Ethnicity: Not reported Age: VBE: Mean = 25 (SD ± 3.2) pCS: Mean = 25 (SD ± 3.4) Parity: 100% primiparous	Cross sectional	How: hysician-administered questionnaire in interview format Inclusion rate: 85.5% (213/249)	6 to 24 months postpartum	Number: *n* = 213 Recruitment: From June 2009 until September 2010, women who had been referred to health clinics at Tehran University of Medical Sciences were recruited.	VBE: 53.5% (*n* = 114) pCS: 46.5% (*n* = 99)	FSFI total score FSFI domain scores	Analysis: Student’s two tailed *t* test, Chi square test and one-way ANOVA Multivariable regression model: To adjust for confounders (age, duration of marriage, educational level, contraceptive method and job)	No significant differences in total FSFI or FSFI domain scores were found between VBE and pCS groups.
Mohammed et al. 2016 [76]	Country: Egypt Ethnicity: Not reported Age: 18–35 (range) Parity: Not reported	Cross sectional	How: Questionnaire Inclusion rate: Not reported	12 months postpartum	Number: *n* = 300 Recruitment: Females attending the Dermatology & Venereology Outpatient Clinic of Al-Azhar University Hospitals and the Family Planning Clinic of Zagazig General Hospital	SVB: 17.3% (*n* = 52) VBE: 34.7% (*n* = 104) AVB: 2.0% (*n* = 6) pCS: 40.7% (*n* = 122) eCS: 5.3% (*n* = 16)	FSFI total score FSFI domain scores.	Analysis: Chi-square test, paired *t* test and ANOVA were used	Mean score for FSFI domain desire was significantly worse at one year postpartum in AVB group (mean = 3.0, SD ± 0.0) compared to other mode of birth groups (SVB mean = 3.3, SVB SD ± 0.42; VBE mean = 3.6, VBE SD ± 0.4; pCS mean = 3.6, pCS SD ± 0.4; eCS mean = 3.9, eCS SD ± 0.4), *p* ≤ 0.001 Mean score for FSFI domain arousal was significantly worse at 1 year postpartum in AVB group (mean = 0.5, SD ± 0.3) compared to other mode of birth groups (SVB mean = 3.6, SVB SD ± 0.2; VBE mean = 3.8, VBE SD ± 0.4; pCS mean = 3.4, pCS SD ± 0.5; eCS mean = 4.2, eCS SD ± 0.4), *p* ≤ 0.001 Mean score for FSFI domain orgasm was significantly worse at 1 year postpartum in SVB group (mean = 3.9, SD ± 0.6) compared to other mode of birth groups (VBE mean = 4.3, VBE SD ± 0.5; AVB mean = 4.5, AVB SD ± 0.2; pCS mean = 4.3, pCS SD ± 0.6; eCS mean = 4.2, eCS SD ± 0.3), *p* ≤ 0.001 Mean score for FSFI domain satisfaction was significantly worse at 1 year postpartum for pCS group (mean = 3.5, SD ± 0.4) compared to other mode of birth groups (SVB mean = 3.9, SVB SD ± 0.6; VBE mean = 3.7, VBE SD ± 0.6; AVB mean = 3.7, AVB SD ± 0.2; eCS mean = 3.8, eCS SD ± 0.7), *p* = 0.003. Mean score for FSFI domain pain was significantly worse at 1 year postpartum in AVB group (mean = 2.9, SD ± 0.3) compared to other mode of birth groups (SVB mean = 4.3, SVB SD ± 0.7; VBE mean = 3.4, VBE SD ± 0.6; pCS mean = 3.6, pCS SD ± 0.6; eCS mean = 3.6, eCS SD ± 0.8), *p* ≤ 0.001 No significant difference for an total mean FSFI score between modes of birth groups.
Saydam et al. 2017 [63]	Country: Turkey Ethnicity: Not reported Age: Not reported Parity: 50.7% primiparous; 49.3% multiparous	Cross sectional	How: interview Inclusion rate: 78.4% (142/181)	6–12 months postpartum	Number: *n* = 142 Recruitment: Women who gave birth in past year who were registered at the Public Health Training and Research Centers (75. Yil, Adalet and Postacilar) in Izmir.	SVB: Total: 45.8% (*n* = 65) Primiparous: 21.8% Multiparous: 23.9% CS: Total: 54.2% (*n* = 77) Primiparous: 28.9% Multiparous: 25.4%	FSFI total score FSFI domain scores FSFI dichotomous score: scores ≤26.55 are considered as female sexual dysfunction	Analysis: Chi-square test, independent sample *t*-test and logistic regression analysis	No significant difference in total or domain specific FSFI scores between SVB and CS groups.
Zgliczynska et al., 2021 [77]	Country: Poland Ethnicity: Not reported Age: Mean = 27.1 (SD ± 4.4); 18–39 (range) Parity: 64.9% primiparous; 35.1% multiparous	Cross sectional	How: online survey Inclusion rate: 53.3% (231/433)	T3: 7–9 months postpartum T4: 10–12 months postpartum	Number: *n* = 433 Recruitment: Websites and forums dedicated to motherhood, pregnancy and childbirth between June–November 2018	SVB: 16.2% VB with perineal trauma (VBE/perineal tear/ AVB): 47.8% CS: 36.0%	FSFI total score FSFI dichotomous score: sexual dysfunction was defined with a score <26.55	Analysis: Mann–Whitney U test, ANOVA Kruskal–Wallis one way analysis of variance, post hoc Dunn’s test of multiple comparisons, Wilcoxon signed-rank, crosstabulation and Chi-squared test. Multiple logistic regression: Univariate analyses and multiple logistic regression model using backward stepwise construction were used	Median FSFI score for arousal (FSFI domain) was significantly worse at 7–9 months in CS (median = 4.5, IQR = 3.3, 5.1) compared to SVB (median = 5.1, IQR = 3.9, 5.7) and VB with perineal tear (median = 4.8, IQR = 4.2, 5.7), *p* = 0.032. Increased risk of female sexual dysfunction in first year after birth in vaginal labour with vacuum/forceps (OR = 3.98, 95%CI = 1.14–13.97, *p* = 0.031). No significant differences for total FSFI score (median) found between mode of birth groups at 10–12 months postpartum. After adjustment in multivariate analysis: No significant association between mode of birth and sexual dysfunction
Baytur et al. 2005 [59]	Country: Turkey Ethnicity: Not reported Age: VBE: Mean = 31.0, (SD ± 5.29) pCS: Mean = 29.1 (SD ± 4.4) Parity: VBE: 80% primiparous; 20% multiparous pCS: 76% primiparous; 24% multiparous	Case control	How: In person interview. Inclusion rate: 47% (53/100)	≥2 years postpartum	Number: *n* = 53 + 15 controls Recruitment: Cases: Women who gave birth in Celal Bayar University School of Medicine Obstetrics Department were randomly selected from obstetric records. Controls: 15 nulliparous attending gynaecology clinics with other symptoms agreed to participate.	VBE: 60.4% (*n* = 32) pCS: 39.6% (*n* = 21) No birth (control): *n* = 15	FSFI total score FSFI domain scores	Analysis: The Mann–Whitney U-test and the Chi-square test	No significant difference in total and domain specific FSFI scores were found between pCS and VBE
Cai et al. 2014 [84]	Country: China Ethnicity: Not reported Age: VBE: Mean = 36.7 (SD ± 5.2) CS: Mean = 34.5 (SD ± 3.8) Parity: 100% primiparous	Case-control	How: Interview questionnaire Inclusion rate: Not reported	At least 1 year since birth	Number: *n* = 165 How: Questionnaire given during initial visit to Department of Infertility and Sexual Medicine, The Third Affiliated Hospital, Sun Yat-sen University for fertility issues, from January 2011 to May 2011.	VBE: 53.3% (*n* = 88) CS: 46.7% (*n* = 77)	FSFI total score FSFI domain scores	Analysis: Chi-square test and *t*-test	No significant differences in total FSFI scores or domain specific FSFI scores between mode of birth groups.
Doğan et al. 2017 [60]	Country: Turkey Ethnicity: Not reported Age: Mean = 30.4 (SD ± 2.4) Parity: 100% primiparous	Case control	How: Face to face interview Inclusion rate: Not reported	≥5 years postpartum	Number: *n* = 150 Recruitment: Selected from those who admitted to outpatient clinic of GATA Haydarpasa Training Hospital for routine control or with fertility desire in the period between January and July 2015.	VBE: 33.3% (*n* = 50) pCS: 33.3% (*n* = 50) No birth (control): 33.3% (*n* = 50)	FSFI total score FSFI categorised scores for measure of sexual dysfunction: 0–15: severe; 16–25: moderate; 26–35: mild; ≥36: normal	Analysis: One-way ANOVA and Chi-square test were used	Mean total FSFI score was significantly worse for VBE group (mean = 65.1, SD ± 10.2) compared to pCS group (mean = 70.5, SD ± 13.3) and control group (mean = 73.4, SD ± 8.8). *p* ≤ 0.001 Mean FSFI score for sexual desire domain was significantly worse for VBE group (mean = 6.1, SD ± 1.7) compared to control group (mean = 7.2, SD ± 1.5). *p* ≤ 0.01 Mean FSFI score for arousal domain was significantly worse for VBE group (mean = 12.4, SD ± 2.3) compared to pCS group (mean = 14.8, SD ± 2.8) and control group (mean = 15.7, SD ± 2.5). *p* ≤ 0.001 Mean FSFI score for orgasm domain was significantly worse for VBE group (mean = 10.6, SD ± 2.5) compared to pCS group (mean = 11.8, SD ± 2.6) and control group (mean = 12.4, SD ± 1.9). *p* ≤ 0.001 No significant differences in FSFI total score or FSFI domain specific scores between pCS and control groups. No groups suffered from any degree of sexual dysfunction.
Crane et al. 2013 [65]	Country: USA Ethnicity: Not reported Age: AVB: Mean = 28.8 (SD ± 5.7) CS: Mean = 29.5 (SD ± 5.0) Parity: 100% primiparous	Retrospective cohort	How: Telephone interview Inclusion rate: 32.3% (109/337)	1 year postpartum	Number: *n* = 109 Recruitment: Women with second-stage arrest in first pregnancy who delivered between January 2009–May 2011 at 2 different institutions were identified by obstetric database using codes.	AVB: 48.6% (*n* = 53) pCS: 33% (*n* = 36) CS after failed AVB: 18.3% (*n* = 20)	FSFI total score FSFI domain scores FSFI dichotomous score: sexual dysfunction was defined with a score < 26.55	Analysis: Chi-square test, Fisher exact test for categorical data and student *t* test for continuous data	Mean FSFI score for orgasm domain was significantly better in pCS group (mean = 4.52, SD not reported) compared to AVB group (mean = 3.78, SD not reported), *p* = 0.05. Mean FSFI score for overall sexual satisfaction was significantly better in pCS group (mean = 4.70, SD not reported) compared to AVB group (mean = 4.16, SD not reported), *p* = 0.04. No significant difference in amount of sexual dysfunction between AVB and pCS groups.
Fehniger et al. 2013 [66]	Country: USA Ethnicity: White/Caucasian: 48% (*n* = 526) Black/African American: 20% (*n* = 214) Asian/Asian-American: 16% (171) Latina/Hispanic: 17% (*n* = 183) Age (at time of questionnaire ≥10 years postpartum): Mean = 56.3 (SD ± 8.7) Parity: 12% primiparous; 88% multiparous	Retrospective cohort	How: Self-administered questionnaire Inclusion rate: 59.3% (1094/1844)	95%: ≥10 years postpartum	Number: *n* = 1094 Recruitment: Long-term enrolees in the Kaiser Permanente Northern California (KPNC) who were participating in Reproductive Risks of Incontinence Study at Kaiser 2, January 2003 to January 2008.	SVB/VBE/AVB: 86% (*n* = 945) CS: 7% (*n* = 75)	FSFI domains (modified)	Multivariable logistic regression: Models adjusted for age, race/ethnicity, partner status, self-reported general health, diabetes status.	After adjustment: Increased risk of low sexual desire or interest >= 10 years postpartum in those who underwent AVB, compared to those who did not (aOR = 1.38, 95% CI = 1.04–1.83, *p* < 0.05).
Gungor et al. 2007 [61]	Country: Turkey Ethnicity: Not reported Age: SVB/VBE/AVB: Mean = 27.9 (SD ± 3.3) CS: Mean = 28.5 (SD ± 3.9) Parity: 100% primiparous	Retrospective cohort	How: Self-reported questionnaire at clinic. Inclusion rate: 90% (135/150)	≥12 months postpartum Time since birth (years): VD/ VBE/ AVB: Mean = 5.6, SD ± 3.5 CS: Mean = 4.8, SD ± 2.7	Number: *n* = 135 Recruitment: From December 2005 until March 2006, primiparous women with a prior live birth were identified upon their presentation at the outpatient desk and were asked to participate.	SVB/ VBE/ AVB: 66.6% (*n* = 90) CS: 33.3% (*n* = 45)	GRISS overall sexual function score GRISS domain scores: 1–4: degrees of sexual satisfaction 5–9: degrees of sexual dissatisfaction	Analysis: Chi-square test and Mann–Whitney U-test were used	No significant differences in total or domain specific GRISS scores between SVB/VBE/AVB and CS groups.
Klein et al. 2015 [75]	Country: Austria Ethnicity: Caucasian Age: VD: Mean = 29.6 (SD ± 5.68) CS: Mean = 33.3 (SD ± 5.24) Parity: 100% primiparous	Retrospective cohort	How: Questionnaire Inclusion rate: 63.9% (99/155)	12–18 months postpartum	Number: *n* = 99 Recruitment: All primiparous women who delivered consecutively at the Department of Obstetrics and Gynecology of the Medical University of Vienna in a period of 10 months were identified using the birth register.	SVB: 56% (*n* = 55) pCS: 44% (*n* = 44)	FSFI total score FSFI domain scores	Analysis: Mann–Whitney U-test Multivariate regression analysis: To control for confounders (BMI, age, education)	No significant differences in total or domain specific FSFI scores between SVB and CS groups.
Song et al. 2014 [79]	Country: Japan Ethnicity: Not reported Age: Mean = 33.2 (SD ± 4.4) Parity: Not reported	Retrospective cohort	How: Postal questionnaire Inclusion rate: 86.7% (435/502)	6 months postpartum	Number: *n* = 435 Recruitment: From November 2011 until June 2013, mothers were recruited during postnatal examination at Kawasaki University Hospital and related hospitals.	SVB/ VBE: 83.7% (*n* = 364) AVB: 5% (*n* = 21) pCS: 5% (*n* = 23) eCS: 6% (*n* = 27)	SFQ28 domain scores	Analysis: ANOVA, Wilcoxon signed-rank test and Spearman rank analysis. Multivariable regression analysis: To control for confounders (age).	Mean SFQ8 score for partner domain was significantly worse in SVB/VBE group (mean = 8.7, SD ± 2.2) compared to AVB group (mean = 9.1, SD ± 1.5). After adjustment in multivariable regression analysis: Caesarean section was a significant predictor of the domains of desire (*p* = 0.015), arousal (lubrication) (*p* = 0.032), and arousal (cognitive) (*p* = 0.036).
Baksu et al. 2006 [64]	Country: Turkey Ethnicity: Not reported Age: 15–35 (range) Parity: 100% primiparous	Prospective cohort	How: In person questionnaire Inclusion rate: 80.5% (248/308)	T1: 6 months postpartum	Number: *n* = 248 Recruitment: Primiparous women who applied to antenatal clinic from January–July 2004 and gave birth in clinic.	VBE: 62.9% (*n* = 156) pCS: 37.1% (*n* = 92)	FSFI total score FSFI domain scores	Analysis: *t* test was used for comparison between mode of birth groups.	At 6 months, VBE had a significantly worse mean total FSFI score (mean = 22.16, SD ± 3.68) compared to pCS (mean = 28.32, SD ± 6.53), *p* = ≤ 0.001 At 6 months, VBE had a significantly worse mean FSFI score for FSFI domains arousal (VBE = 3.75 ± 0.86, pCS = 4.83 ± 0.64), lubrication (VBE = 4.2 ± 0.3, pCS = 5.04 ± 0.67), orgasm (VBE = 3.36 ± 0.37, pCS = 5.48 ± 0.48), satisfaction (VBE = 4.92 ± 0.31, pCS = 5.5 ± 0.52), and pain (VBE = 3.12 ± 0.32, pCS = 4.8 ± 0.7) when compared to pCS. *p* ≤ 0.001
Chang et al. 2015 [85]	Country: Taiwan Ethnicity: Not reported Age: VB: Mean = 32.1 (SD ± 0.3) CS: Mean = 33.9 (SD ± 0.3) Parity: VB: 53.5% primiparous CS: 53.6% primiparous	Prospective cohort	How: Not reported Inclusion rate: 47.7% (351/736)	T3: 6 months postpartum T4: 12 months postpartum	Number: *n* = 351 Recruitment: When they gave birth in maternity unit of a medical centre in Taiwan, between 2010 and 2011.	VB: 57.0% (*n* = 200) CS: 43.0% (*n* = 151)	FSFI total score FSFI domain scores High FSFI values were defined with a score >26.55	Analysis: Differences in demographic characteristics explored with MANOVA. MANCOVA: Comparisons of sexual function between vaginal and caesarean groups, adjusting for significant differences in demographic covariates. Covariates included: gestational age, personal income, age, current body mass index (BMI), personal income, feeding type, gestational age of the baby, and neonatal care unit attended.	No significant differences in total FSFI scores, domain specific FSFI scores or rate of high FSFI scores between mode of birth groups.
Dahlgren et al. 2022 [86]	Country: Sweden Ethnicity: Not reported Age: <25: 11.27% (*n* = 79) 25–29: 48.2% (*n* = 338) 30–34: 32.1% (*n* = 225) ≥35: 8.4% (*n* = 59) Parity: 100% primiparous	Prospective cohort	How: Web-based questionnaires Inclusion rate: 66.8% (701/958)	T2: 12 months postpartum	Number: *n* = 701 Recruitment: Nulliparous women registering for maternity health care in the region Örebro County, Sweden, were invited to participate, from 1 October 2014 to 1 October 2017.	SVB: 69.0% (*n* = 658) AVB: 15.5% (*n* = 148) CS: 15.5% (*n* = 148)	PISQ-12 total score PISQ-12 sub scores (satisfaction-question 4 of questionnaire; dyspareunia- question 5 of questionnaire)	Generalised estimating equations: Coefficients and odds ratios obtained using GEEs. Adjusted estimates controlled for age, education, smoking, BMI, self-reported health, delivery mode, episiotomy, mode of start of delivery, gestational age, degree of perineal tear, high vaginal tear and breastfeeding 12 months postpartum.	AVB increased the odds of dyspareunia when compared to SVB (OR = 1.86, 95% CI = 1.19, 2.89). No significant difference in total PISQ-12 score between mode of birth groups. No significant difference in satisfaction PISQ-12 sub-score between mode of birth groups. After adjustment: Association between AVB and increased odds of dyspareunia no longer significant. After alternative adjusted analysis (excluded perineal tear, high vaginal tear, episiotomy, mode of delivery start): AVB increased the odds of dyspareunia when compared to SVB (OR = 1.63, 95% CI = 1.01, 2.63).
de Sousa et al., 2021 [88]	Country: Portugal Ethnicity: Not reported Age: SVB: 26–36 (range); median = 32 AVB: 28–34 (range); median = 32 Parity: SVB: 46.2% primiparous; 53.8% multiparous AVB: 84.7% primiparous; 15.3% multiparous	Prospective cohort	How: Online survey Inclusion rate: T2: 37% (72/196) T3: 52% (102/196)	T2: 6 months postpartum T3: 12 months postpartum	Number: *n* = 196 Recruitment: Women 3 days postpartum from giving birth at Department of Obstetrics and Gynacologie Hospital de Braga in period February–October 2018	SVB: 33.16% AVB: 66.84%	FSFI total score FSFI dichotomous score: sexual dysfunction was defined with a score <26.55	Analysis: Chi square and Fischer exact test used for categorical variables; independent *t*-test used for continuous variables.	Overall rate of sexual dysfunction at 6 months was 43%. Overall rate of sexual dysfunction at 12 months was 48%. At 12 months, pain FSFI domain was significantly worse in women with AVB (mean = 4.4, SD ± 1.4) when compared to SVB (mean = 5.1, SD ± 1.1). At 6 months, there was no differences in FSFI total score or domains between mode of birth groups. When exploring influence of different types of instruments used in operative vaginal birth (Thierry’s Spatulas or Kiwi Vacuum Extractor), FSFI scores did not differ between the two instruments at 6 or 12 months postpartum
de Souza et al. 2015 [74]	Country: Australia Ethnicity: Not reported Age: Mean = 30.70 (SD not reported) Parity: 100% primiparous	Prospective cohort	How: Postal questionnaire Inclusion rate: T1: 74% (322/437) T2: 65% (286/400)	T1: 6 months postpartum T2: 12 months postpartum	Number *n* = 286 Recruitment: From antenatal clinic Between January 2010 and July 2011.	SVB/VBE: 51.2% (*n* = 133) AVB: 21.9% (*n* = 57) CS: 26.9% (*n* = 70)	FSFI total score. FSFI domain scores.	Analysis: Chi-square test and Student’s *t* test Linear mixed modelling (LMM): Control for confounding factors: maternal age, maternal BMI, gestation at birth, neonatal weight, breastfeeding status and risk of postnatal depression.	No significant differences in FSFI total score or FSFI domain specific scores between mode of birth groups at any time point.
de Souza et al. 2013 [89]	Country: Not reported Ethnicity: Not reported Age: Not reported Parity: 100% primiparous	Prospective cohort	How: Not reported Inclusion rate: 65% (286/440)	T1: 6 months postpartum T2: 12 months postpartum	Number: *n* = 286 Recruitment: Not reported	SVB; 52.3% AVB: 21.3% CS: 26.3% VBE: Of all vaginal births, 37.9% were VBE	FSFI total score. FSFI domain scores.	Analysis: Mann–Whitney or Kruskal–Wallis tests.	At 6 months postpartum, women who had a VBE had significantly worse total FSFI score, significantly worse scores for lubrication and orgasm, and significantly better scores for pain when compared to CS. No significant differences in FSFI total score or FSFI domain specific scores between SVB, AVB or CS groups at 12 months postpartum.
Di Dedda et al. 2017 [81]	Country: Italy Ethnicity: Not reported Age: 18–44 (range); median = 33 Parity: 59.9% primiparous 40.1% multiparous	Prospective cohort	How: Questionnaire Inclusion rate: 52.4% (210/401)	T1: 6 months postpartum	Number: *n* = 210 Recruitment: Women who underwent VD or CS from single urogynaecological unit, between February and July 2016	SVB/ VBE/ AVB: Not reported CS: Not reported	FSFI total score FSFI domain scores	Not reported	Women who underwent CS had significantly better total FSFI score compared to women who underwent SVB/VBE/AVB. *p* ≤ 0.001 (mean scores and SD not reported). Women who underwent CS had significantly better score for FSFI domain pain compared to women who underwent SVB/VBE/AVB. *p* ≤ 0.001 (mean scores and SD not reported).
Kahramanoglu et al. 2017 [62]	Country: Turkey Ethnicity: Not reported Age: VBE: Mean = 23.6 (SD ± 1.2) CS: Mean = 23.7 (SD ± 1.1) Parity: 100% primiparous	Prospective cohort	How: T1: Face to face interview T2 & T3: Telephone interview Inclusion rate: T1: 83.6% (337/403) T2: 74.4% (300/403) T3: 54.1% (218/403)	T1: 6 months postpartum T2: 12 months postpartum T3: 24 months postpartum	Number: T1: *n* = 337 T2: *n* = 300 T3: *n* = 218 Recruitment: All nulliparous women aged 18–45 years in stable relationships who presented to Obstetrics and Gynecology Department, Suleymaniye Women’s Health Training and Research Hospital, Istanbul, Turkey, from February 2012–July 2014.	VBE: 64% (T1) pCS: 36% (T1)	FSFI total score FSFI domain scores	Analysis: Student’s *t* test testing differences in FSFI between MOD in each timepoint.	No significant differences in total or domain specific FSFI scores between VBE and CS groups at any time point.
Lurie et al. 2013 [87]	Country: Israel Ethnicity: Not reported Age: SVB: Mean = 27.7 (SD ± 4.8) VBE: Mean = 28.4 (SD ± 3.9) AVB: Mean = 27.3 (SD ± 5.7) pCS: Mean = 31.2 (SD ± 5.6) eCS: Mean = 29.4 (SD ± 5.4) Parity: SVB: Mean parity = 1.0 (SD ± 0.0) VBE: Mean parity = 1.0 (SD ± 0.0) AVB: Mean parity = 1.0 (SD ± 0.0) pCS: Mean parity = 1.1 (SD ± 0.3) eCS: Mean parity = 1.1 (SD ± 0.2)	Prospective cohort	How: Telephone questionnaire Inclusion rate: 82% (82/100)	T3: 6 months postpartum	Number: *n* = 82 Recruitment: Women approached on day of discharge from Maternity Ward of Edith Wolfson Medical Centre between January 2010 and February 2011.	SVB: 19.5% (*n* = 16) VBE: 17.1% (*n* = 14) AVB: 17.1% (*n* = 14) pCS: 20.7% (*n* = 17) eCS: 22.0% (*n* = 18)	FSFI total score FSFI domain scores	Analysis: FSFI scores were compared across delivery type using Kruskal–Wallis test. General linear modelling: Used to assess across-group differences	No significant differences in total or domain specific FSFI scores between mode of birth groups.
Moghadam et al. 2019 [71]	Country: Iran Ethnicity: Not reported Age: Mean = 27.2 (SD ± 4.52) Parity: 100% primiparous	Prospective cohort	How: Questionnaire conducted by midwifery experts. Inclusion rate: 78% (107/136)	T1: 6 months postpartum T2: 12 months postpartum	Number: T2: *n* = 107 Recruitment: Nulliparous mothers aged 18–35 years who referred to 5 randomly selected healthcare centres in Sanandaj during 2015–2016 for infant vaccination.	VBE: 45.8% (*n* = 49) CS: 54.2% (*n* = 58)	FSFI total score FSFI domain scores	Analysis: Interferential and descriptive statistics and paired *t* test. Multiple logistic regression: To adjust for confounders (age, education, job, income, infant weight, sex per week and contraceptive method)	Mean total FSFI score at 6 months postpartum was significantly worse in VBE group (mean = 24.7, SD ± 4.45) compared to CS group (mean = 26.8, SD ± 4.24), *p* = 0.01. Mean score for FSFI domain sexual arousal at 6 months was significantly worse in VBE group (mean = 3.6, SD ± 0.96) compared to CS group (mean = 4.0, SD ± 0.98), *p* = 0.02. Mean score for FSFI domain sexual desire at 6 months was significantly worse in VBE group (mean = 3.4, SD ± 0.87) compared to CS group (mean = 3.8, SD ± 1.006), *p* = 0.001. Mean score for FSFI domain sexual desire at 12 months was significantly worse in VBE group (mean = 3.4, SD ± 0.92) compared to CS group (mean = 3.8, SD ± 0.87), *p* = 0.03. After adjustment for confounders in multiple logistic regression model: Increased odds of better total FSFI score at 6 months postpartum in CS compared to VBE (OR = 1.14, 95% CI = 1.02–1.27, *p* = 0.01). Increased odds of better sexual desire FSFI score at 6 months postpartum in CS compared to VBE (OR = 2.47, 95% CI = 1.15–5.32, *p* = 0.02).
Rogers et al. 2012 [67]	Country: USA Ethnicity: Not reported Age: Mean = 25.2 (SD ± 5.5) Parity: 100% primiparous	Prospective cohort	How: Questionnaire Inclusion rate: Total: 70% (469/672) VD: 74% (331/448) CS: 62% (138/224)	6 months postpartum	Number: *n* = 469 Recruitment: From a population of midwifery patients. Recruited midwifery patients during pregnancy, and additional cohort who underwent CS.	SVB/ VBE/ AVB: 71% CS: 29%	FSFI total score	Multivariate analysis: Controlling for education, BMI and age differences between groups	After adjustment in multivariate analyses: At 6 months postpartum, those who underwent CS had significantly worse total FSFI score (mean = 26.6, SD ±6.3) compared to those in SVB/VBE/AVB group (mean = 28.5, SD ± 5.4). *p* = 0.004
Spaich et al., 2020 [78]	Country: Germany Ethnicity: Not reported Age: Mean = 31.2 (SD ± 5.0); 18–47 (range) Parity: 47.3% primiparous; 52.7% multiparous	Prospective cohort	How: Questionnaires (not clear on modality Inclusion rate: T1: 66% (344/522) T2: 65% (339/522) T3: 58% (303/522)	T2: 6 months postpartum T3: 12 months postpartum	Number: *n* = 522 Recruitment: During peripartum hospitalisation of women giving birth in period 2013–2018 at University Medical Centre Mannheim	SVB: 50.4% (*n* = 263) VBE: 7.28% (*n* = 38) AVB: 7.9% (*n* = 41) CS: 41.8% (*n* = 218)	FSFI total score SAQ total score	Analysis: One way ANOVAs, chi square test on Fisher’s exact test, nonparametric tests (Kruskal–Wallis, Friedman). When significant group differences were found, variables were entered as possible covariates in subsequent analysis.	No significant differences in FSFI total score or SAQ total score between mode of birth groups at any time point.
Szollosi et al., 2021 [83]	Country: Hungary Ethnicity: Not reported Age: At 6 months postpartum: Mean = 31.8 (SD ± 4.7) At 12 months postpartum: Mean = 31.44 (SD ± 3.6) Parity: At 6 months postpartum: 53.7% primiparous; 46.3% multiparous At 12 months postpartum: 54.7% primiparous; 45.3% multiparous	Prospective cohort	How: Online questionnaires Inclusion rate: 729 invited Inclusion rate: T2: 77.0% (214/278) T3: 34.2% (95/278)	T2: 6 months postpartum T3: 12 months postpartum	Number: 3 months: *n* = 293 6 months: *n* = 214 12 months: *n* = 95 Recruitment: Women giving birth in period June 2018-August 2019 in 3 obstetric institutes in Budapest were personally invited within 3 days postpartum	SVB: 6 months = 14.01% 12 months = 16.84% VBE: 6 months = 26.16% 12 months = 34.73% pCS: 6 months = 27.10% 12 months = 26.31% eCS: 6 months = 19.62% 12 months = 12.63%	FSFI, sexual dysfunction was defined with a dichotomous score <26.55	Analysis: Pearson chi square test and Mann–Whitney U test. Variables that were in significant connection with sexual dysfunction in univariate logistic regression model were examined in multivariate logistic regression model. Multivariate logistic regression modelling: Adjustment made for parity, age, and education level	No significant differences in total FSFI scores between mode of birth groups at any time point. Mode of births were not associated with sexual dysfunction at any time point.
Yee et al. 2013 [68]	Country: USA Ethnicity: Caucasian: 51,9% (*n* = 67) African America: 26,4% (*n* = 34) Latina: 7,8% (*n* = 10) Asian: 13,2% (*n* = 17) Other: 0,8% (*n* = 1) Age: Mean = 32.1 (SD ± 6.0) Parity: 34.9% primiparous; 65.1% multiparous	Prospective cohort	How: Telephone & in person questionnaire Inclusion rate: 80.6% (129/160)	T1: 6–8 months postpartum	Number: *n* = 129 Recruitment: From August 2008 to March 2011, letters of invitation with opt-in/opt-out cards were sent to all pregnant women obtaining prenatal care at the University of California, San Francisco obstetrics practice.	SVB/ VBE/ AVB: 76.7% (*n* = 99) CS: 23.3% (*n* = 30)	Modified SHOW-Q	Analysis: Chi-square and *t* test were used Multiple linear regression: To identify predictors of postpartum sexual function (age, race, education, marital status, parity, breastfeeding status and depression)	No significant differences in mean SHOW-Q scores between CS and SVB/VBE/AVB groups.

SVB = spontaneous vaginal birth; VBE = vaginal birth with episiotomy; AVB = assisted vaginal birth; pCS = planned (elective) caesarean section; eCS = emergency caesarean section; VB = vaginal birth (unspecified); CS = caesarean section (unspecified).

**Table 2 ijerph-20-05252-t002:** Quality assessment using NOS.

Study	Selection				Comparability		Outcome			Overall Rating
	Representativeness of Exposed Cohort	Selection of the Non-Exposed Cohort from Same Source as Exposed Cohort	Ascertainment of Exposure	Outcome of Interest Was Not Present at Start of Study	Comparison of Modes of Birth including Specification, e.g., Differentiation between Types of Vaginal Birth (Spontaneous Vaginal Birth, Vaginal Birth with Episiotomy, Assisted Vaginal Birth) and Types of Caesarean Section (Planned, Emergency)	Controls for Any Additional Factor (Demographics, Health, etc.)	Assessment of Outcome	Follow-Up Long Enough for Outcome to Occur	Adequacy of Follow-Up (Rate of Women included in Analysis at Follow Up Point)	
Baksu et al. 2006 [64]	Women who applied to antenatal clinic are likely to be representative of pregnant women ★	All women came from the same source ★	Mode of birth details were ascertained from hospital records after birth ★	Pre-birth sexual function was measured ★	Modes of birth compared and specified ★	Additional factors were controlled for in study design or analysis ★	Self-reported questionnaire ★	Follow up at ≥6 months postpartum ★	Follow up rate was 80.5%. Small number of participants lost unlikely to introduce bias ★	Good
Barbara et al. 2016 [80]	Women recruited before discharge from Department of Women’s and Children’s Health are likely to representative of women who have given birth ★	All women came from the same source ★	Mode of birth was retrieved from hospital records ★	Pre-birth sexual function was measured ★	Comparison of modes of birth but no specific differentiation between spontaneous vaginal birth and vaginal birth with episiotomy.	Additional factors were controlled for in study design or analysis ★	Self-reported questionnaire ★	Follow up at ≥6 months postpartum ★	Follow up rate <80% but description of those lost was provided ★	Good
Baud et al., 2020 [82]	Women whose details were in obstetric database are likely to be representative of women who have given birth ★	All women came from the same source ★	Mode of birth data retrieved from obstetric database, which was collected from hospital records ★	Pre-birth sexual function was not measured and women with sexual dysfunction prior to birth were not excluded	Modes of birth compared and specified ★	Additional factors were controlled for in study design or analysis ★	Self-reported questionnaire ★	Follow up at ≥6 months postpartum ★	Follow up rate <80% and no description of those lost was provided	Good
Baytur et al. 2005 [59]	Women randomly selected from obstetric records are likely to be representative of women who have given birth ★	All women came from the same source ★	Mode of birth data retrieved from obstetric records ★	Pre-birth sexual function was not measured and women with sexual dysfunction prior to birth were not excluded	Modes of birth compared and specified ★	Additional factors were controlled for in study design or analysis ★	Self-reported questionnaire ★	Follow up at ≥6 months postpartum ★	Follow up rate <80% and no description of those lost was provided	Good
Cai et al. 2014 [84]	Women who attended a clinic for fertility issues are somewhat representative of women who have given birth ★	All women came from the same source ★	No description of how mode of birth data was ascertained	Pre-birth sexual function was not measured and women with sexual dysfunction prior to birth were not excluded	Comparison of modes of birth but no specific differentiation between planned caesarean section and emergency caesarean section.	Additional factors were controlled for in study design or analysis ★	Self-reported questionnaire ★	Follow up at ≥6 months postpartum ★	Follow up rate not stated	Fair
Chang et al. 2015 [85]	Women who gave birth in a maternity unit are likely to be representative of women who have given birth ★	All women came from the same source ★	Mode of birth data retrieved from medical records ★	Pre-birth sexual function was not measured and women with sexual dysfunction prior to birth were not excluded	Comparison of modes of birth but no specific differentiation between different types of vaginal birth (spontaneous, with episiotomy or assisted) and caesarean section (planned or emergency)	Additional factors were controlled for in study design or analysis ★	Self-reported questionnaire ★	Follow up at ≥6 months postpartum ★	Follow up rate <80% but description of those lost was provided ★	Good
Crane et al. 2013 [65]	Women identified in obstetric database are likely to be representative of women who have given birth ★	All women came from the same source ★	Mode of birth data retrieved from online medical records ★	Pre-birth sexual function was not measured and women with sexual dysfunction prior to birth were not excluded	Modes of birth compared and specified ★	Additional factors were controlled for in study design or analysis ★	Self-reported questionnaire ★	Follow up at ≥6 months postpartum ★	Follow up rate <80% and no description of those lost was provided	Good
Dabiri et al. 2014 [69]	Women who brought children for vaccination are likely to be representative of women who have given birth ★	All women came from the same source ★	Mode of birth data was self-reported by patients.	Women with sexual dysfunction prior to pregnancy were excluded ★	Modes of birth compared and specified ★	Additional factors were controlled for in study design or analysis ★	Self-reported questionnaire ★	Follow up at ≥6 months postpartum ★	Follow up rate not stated	Good
Dahlgren et al. 2022 [86]	Women registering for maternity health care are likely to be representative of women who have given birth ★	All women came from the same source ★	Mode of birth data retrieved from online database ★	Pre-birth sexual function was measured ★	Comparison of modes of birth but no specific differentiation between planned caesarean section and emergency caesarean section.	Additional factors were controlled for in study design or analysis ★	Self-reported questionnaire ★	Follow up at ≥6 months postpartum ★	Follow up rate <80% and no description of those lost was provided	Good
De Sousa et al. 2021 [88]	Women who gave birth in an obstetric and gynaecology department are likely to representative of women who have given birth ★	All women came from the same source ★	Mode of birth was ascertained from medical records ★	Pre-birth sexual function was not measured and women with sexual dysfunction prior to birth were not excluded	Modes of birth compared and specified ★	Additional factors were controlled for in study design or analysis ★	Self-reported questionnaire ★	Follow up at ≥6 months postpartum ★	Follow up rate <80% and no description of those lost was provided	Good
De Souza et al. 2015 [74]	Women who presented to an antenatal clinic are likely to be representative of women who have given birth ★	All women came from the same source ★	Mode of birth was ascertained from medical records ★	Pre-birth sexual function was measured ★	Comparison of modes of birth but no specific differentiation between different types of vaginal birth (spontaneous or with episiotomy) and caesarean section (planned or emergency)	Additional factors were controlled for in study design or analysis ★	Self-reported questionnaire ★	Follow up at ≥6 months postpartum ★	Follow up rate <80% and no description of those lost was provided	Good
De Souza et al. 2013 [89]	No description of the derivation of the cohort	All women came from the same source ★	Mode of birth was ascertained from hospital database ★	Pre-birth sexual function was measured ★	Comparison of modes of birth but no specific differentiation between planned caesarean section and emergency caesarean section.	Additional factors were not controlled for in study design or analysis.	Self-reported questionnaire ★	Follow up at ≥6 months postpartum ★	Follow up rate <80% and no description of those lost was provided	Poor
Dean et al. 2008 [73]	Women who gave birth in one of 3 different maternity units are likely to be representative of women who have given birth ★	All women came from the same source ★	Mode of birth was ascertained from medical records ★	Pre-birth sexual function was not measured and women with sexual dysfunction prior to birth were not excluded	Comparison of modes of birth but no specific differentiation between planned caesarean section and emergency caesarean section.	Additional factors were controlled for in study design or analysis ★	Self-reported questionnaire ★	Follow up at ≥6 months postpartum ★	Follow up rate <80% but description of those lost was provided ★	Good
Di Dedda et al. 2017 [81]	Women who gave birth in a urogynaecological unit are likely to be representative of women who have given birth ★	All women came from the same source ★	Mode of birth data was self-reported by patients.	Women with sexual dysfunction prior to birth were excluded ★	Comparison of modes of birth but no specific differentiation between different types of vaginal birth (spontaneous, with episiotomy or assisted) and caesarean section (planned or emergency)	Additional factors were controlled for in study design or analysis ★	Self-reported questionnaire ★	Follow up at ≥6 months postpartum ★	Follow up rate <80% and no description of those lost was provided	Good
Doğan et al. 2016 [60]	Women who presented to a clinic with fertility desire are somewhat representative of women who have given birth ★	All women came from the same source ★	No description of how mode of birth data was ascertained	Pre-birth sexual function was not measured and women with sexual dysfunction prior to birth were not excluded	Modes of birth compared and specified ★	Additional factors were controlled for in study design or analysis ★	Self-reported questionnaire ★	Follow up at ≥6 months postpartum ★	Follow up rate not stated	Fair
Fehniger et al. 2013 [66]	Enrolees in study who were participating in a reproductive study are somewhat representative of women who have given birth ★	All women came from the same source ★	Mode of birth was ascertained from medical records ★	Pre-birth sexual function was not measured and women with sexual dysfunction prior to birth were not excluded	Comparison of modes of birth but no specific differentiation between different types of vaginal birth (spontaneous or with episiotomy) and caesarean section (planned or emergency)	Additional factors were controlled for in study design or analysis ★	Self-reported questionnaire ★	Follow up at ≥6 months postpartum ★	Follow up rate <80% and no description of those lost was provided	Good
Ghorat et al. 2017 [70]	A random sample of women who had given birth in 1 of many hospitals is representative of women who have given birth ★	All women came from the same source ★	Mode of birth data was self-reported by patients.	Pre-birth sexual function was not measured and women with sexual dysfunction prior to birth were not excluded	Comparison of modes of birth but no specific differentiation between different types of vaginal birth (spontaneous or with episiotomy) and caesarean section (planned or emergency)	Additional factors were controlled for in study design or analysis ★	Self-reported questionnaire ★	Follow up at ≥6 months postpartum ★	Follow up rate not stated	Fair
Gungor et al. 2007 [61]	Women who presented to outpatient desk who had a prior birth are representative of women who have given birth ★	All women came from the same source ★	Mode of birth data was self-reported by patients.	Pre-birth sexual function was not measured and women with sexual dysfunction prior to birth were not excluded	Comparison of modes of birth but no specific differentiation between different types of vaginal birth (spontaneous, with episiotomy or assisted) and caesarean section (planned or emergency)	Additional factors were controlled for in study design or analysis ★	Self-reported questionnaire ★	Follow up at ≥6 months postpartum ★	Follow up rate was 90%. Small number of participants lost unlikely to introduce bias ★	Fair
Hosseini et al. 2012 [72]	Women who had been referred to health clinics are somewhat representative of women who have given birth ★	All women came from the same source ★	Mode of birth was ascertained from structured interview ★	Women with sexual dysfunction prior to birth were excluded ★	Modes of birth compared and specified ★	Additional factors were controlled for in study design or analysis ★	Self-reported questionnaire ★	Follow up at ≥6 months postpartum ★	Follow up rate was 85.5%. Small number of participants lost unlikely to introduce bias ★	Good
Kahramanoglu et al. 2017 [62]	Women in stable relationships who presented to health clinics are representative of women who have given birth ★	All women came from the same source ★	No description of how mode of birth data was ascertained	Pre-birth sexual function was measured ★	Modes of birth compared and specified ★	Additional factors were controlled for in study design or analysis ★	Self-reported questionnaire ★	Follow up at ≥6 months postpartum ★	Follow up rate <80% and no description of those lost was provided	Good
Klein et al. 2015 [75]	Women identified in birth register who gave birth consecutively at a tertiary care centre serving high-risk pregnancies with different pregnancy-associated complications are somewhat representative of women who have given birth ★	All women came from the same source ★	No description of how mode of birth data was ascertained	Women with sexual dysfunction prior to birth were excluded ★	Modes of birth compared and specified ★	Additional factors were controlled for in study design or analysis ★	Self-reported questionnaire ★	Follow up at ≥6 months postpartum ★	Follow up rate <80% and no description of those lost was provided	Good
Lurie et al. 2013 [87]	Women who were discharged after birth in a maternity ward are likely to be representative of women who have given birth ★	All women came from the same source ★	No description of how mode of birth data was ascertained	Pre-birth sexual function was not measured and women with sexual dysfunction prior to birth were not excluded	Modes of birth compared and specified ★	Additional factors were controlled for in study design or analysis ★	Self-reported questionnaire ★	Follow up at ≥6 months postpartum ★	Follow up rate was 82%. Small number of participants lost unlikely to introduce bias ★	Fair
Moghadam et al. 2019 [71]	Mothers referred from 5 randomly selected centres are representative of women who have given birth ★	All women came from the same source ★	Mode of birth data was self-reported by patients.	Pre-birth sexual function was measured ★	Modes of birth compared and specified ★	Additional factors were controlled for in study design or analysis ★	Self-reported questionnaire ★	Follow up at ≥6 months postpartum ★	Follow up rate <80% and no description of those lost was provided	Good
Mohammed et al. 2014 [76]	The married women included in this study are only somewhat representative of women who have given birth because of the high number of women who refused to participate because of their views on discussing their sexual life ★	All women came from the same source ★	No description of how mode of birth data was ascertained	Pre-birth sexual function was measured ★	Modes of birth compared and specified ★	Additional factors were controlled for in study design or analysis ★	Self-reported questionnaire ★	Follow up at ≥6 months postpartum ★	Follow up rate not stated	Good
Rogers et al. 2012 [67]	Midwifery patients recruited during pregnancy are representative of women who had given birth ★	All women came from the same source ★	No description of how mode of birth data was ascertained	Pre-birth sexual function was not measured and women with sexual dysfunction prior to birth were not excluded	Comparison of modes of birth but no specific differentiation between different types of vaginal birth (spontaneous, with episiotomy or assisted) and caesarean section (planned or emergency)	Additional factors were controlled for in study design or analysis ★	Self-reported questionnaire ★	Follow up at ≥6 months postpartum ★	Follow up rate <80% and no description of those lost was provided	Fair
Saydam et al. 2017 [63]	Women who gave birth and are registered at research centres are somewhat representative of women who had given birth ★	All women came from the same source ★	Mode of birth data was self-reported by patients.	Presence of pre-birth sexual dysfunction was ascertained★	Comparison of modes of birth but no specific differentiation between planned caesarean section and emergency caesarean section.	Additional factors were controlled for in study design or analysis ★	Self-reported questionnaire ★	Follow up at ≥6 months postpartum ★	Follow up rate <80% and no description of those lost was provided	Fair
Song et al. 2014 [79]	Women recruited during postnatal examination are representative of women who have given birth ★	All women came from the same source ★	Mode of birth data was self-reported by patients.	Pre-birth sexual function was not measured and women with sexual dysfunction prior to birth were not excluded	Comparison of modes of birth but no specific differentiation between spontaneous vaginal birth and vaginal birth with episiotomy	Additional factors were controlled for in study design or analysis ★	Self-reported questionnaire ★	Follow up at ≥6 months postpartum ★	Follow up rate was 86.7%. Small number of participants lost unlikely to introduce bias ★	Fair
Spaich et al. 2020 [78]	Women recruited during peripartum hospitalisation are somewhat representative of women who have given birth ★	All women came from the same source ★	No description of how mode of birth data was ascertained	Pre-birth sexual function was measured ★	Comparison of modes of birth but no specific differentiation between planned caesarean section and emergency caesarean section.	Additional factors were controlled for in study design or analysis ★	Self-reported questionnaire ★	Follow up at ≥6 months postpartum ★	Follow up rate <80% and no description of those lost was provided	Good
Szollosi et al. 2021 [83]	Women giving birth in obstetric centres are somewhat representative of women who have given birth ★	All women came from the same source ★	Mode of birth data was self-reported by patients.	Pre-birth sexual function was not measured and women with sexual dysfunction prior to birth were not excluded	Modes of birth compared and specified ★	Additional factors were controlled for in study design or analysis ★	Self-reported questionnaire ★	Follow up at ≥6 months postpartum ★	Follow up rate <80% and no description of those lost was provided	Fair
Yee et al. 2013 [68]	Women obtaining prenatal care from hospital are representative of women who have given birth ★	All women came from the same source ★	Mode of birth was ascertained from structured interview ★	Pre-birth sexual function was not measured and women with sexual dysfunction prior to birth were not excluded	Comparison of modes of birth but no specific differentiation between different types of vaginal birth (spontaneous, with episiotomy or assisted) and caesarean section (planned or emergency)	Additional factors were controlled for in study design or analysis ★	Self-reported questionnaire ★	Follow up at ≥6 months postpartum ★	Follow up rate was 80.6%. Small number of participants lost unlikely to introduce bias ★	Good
Zinczenko et al. 2021 [77]	Recruitment from websites dedicated to pregnancy etc are somewhat representative of women giving birth ★	All women came from the same source ★	Mode of birth data was self-reported by patients.	Pre-birth sexual function was measured ★	Comparison of modes of birth but no specific differentiation between different types of vaginal birth (with episiotomy or assisted) and caesarean section (planned or emergency)	Additional factors were controlled for in study design or analysis ★	Self-reported questionnaire ★	Follow up at ≥6 months postpartum ★	Follow up rate <80% and no description of those lost was provided	Good
	Good quality									
	Fair quality									
	Poor quality									

**Table 3 ijerph-20-05252-t003:** Medium-term associations between specific mode of birth groups and sexual function (≥six months and <12 months postpartum).

	SVB vs. VBE	SVB vs. AVB	SVB vs. pCS	SVB vs. eCS	VBE vs. AVB	VBE vs. pCS	VBE vs. eCS	AVB vs. pCS	AVB vs. eCS	pCS vs. eCS
Baksu et al. 2007 [64]						pCS protective ^1,2,4,5,6,7^				
Barbara et al. 2016 [80]								pCS protective ^1,2,4,5^		
Dabiri et al. 2014 [69]										
De Sousa et al. 2021 [88]										
Kahramanoglu et al. 2017 [62]										
Lurie et al. 2013 [87]										
Spaich et al. 2020 [78]										
Szollosi et al. 2021 [83]										
Zgliczynska et al. 2021 [77]		SVB protective ^1^								
	Adjusted association between mode of birth and sexual function									
	Crude association between mode of birth and sexual function									
	No association between mode of birth and sexual function									
	No association calculated between mode of birth and sexual function									

SVB = spontaneous vaginal birth, VBE = vaginal birth with episiotomy, AVB = assisted vaginal birth, pCS = planned caesarean section, eCS = emergency caesarean section; Domains: ^1^ Total sexual function score, ^2^ Arousal, ^4^ Lubrication, ^5^ Orgasm, ^6^ Satisfaction, ^7^ Pain.

**Table 4 ijerph-20-05252-t004:** Longer-term associations between specific mode of birth groups and sexual function (≥12 months postpartum).

	SVB vs. VBE		SVB vs. AVB	SVB vs. pCS	SVB vs. eCS	VBE vs. AVB	VBE vs. pCS	VBE vs. eCS	AVB vs. pCS	AVB vs. eCS	pCS vs. eCS
Baud et al. 2020 [82]				SVB protective ^3,7^							
Crane et al. 2013 [65]									pCS protective ^5,6^		
Dahlgren et al. 2022 [86]			SVB protective ^7^								
De Sousa et al. 2021 [88]			SVB protective ^7^								
Doğan et al. 2016 [60]							pCS protective ^1,2,5^				
Kahramanoglu et al. 2017 [62]											
Klein et al. 2009 [75]											
Mohammed et al. 2014 [76]	SVB protective ^6,7^ VBE protective ^2,3,5^		SVB protective ^2,3,6,7^ AVB protective ^5^	SVB protective ^2,6,7^ pCS protective ^3,5^	SVB protective ^6,7^ eCS protective ^2,3,5^	VBE protective ^2,3,7^ AVB protective ^5^	VBE protective ^2,6^ pCS protective ^7^	VBE protective ^5^ eCS protective ^2,3,6,7^	AVB protective ^5,6^ pCS protective ^2,3,7^	AVB protective ^5^ eCS protective ^2,3,6,7^	pCS protective ^5^ eCS protective ^2,3,6^
Spaich et al. 2020 [78]											
Szollosi et al. 2021 [83]											
		Adjusted association between mode of birth and sexual function									
		Crude association between mode of birth and sexual function									
		No association between mode of birth and sexual function									
		No association calculated between mode of birth and sexual function									

SVB = spontaneous vaginal birth, VBE = vaginal birth with episiotomy, AVB = assisted vaginal birth, pCS = planned caesarean section, eCS = emergency caesarean section; Domains: ^1^ Overall sexual dysfunction, ^2^ Arousal, ^3^ Desire, ^5^ Orgasm, ^6^ Satisfaction, ^7^ Pain.

**Table 5 ijerph-20-05252-t005:** Medium-term associations between combinations of modes of birth and sexual function (≥six months and <12 months postpartum).

	SVB vs. VB ^a^	SVB vs. CS	VBE vs. CS	AVB vs. VB ^b^	AVB vs. CS	VB vs. CS	VB vs. pCS
Barbara et al. 2016 [80]				VB protective ^5^			
Chang et al. 2015 [85]							
De Sousa et al. 2021 [88]							
De Souza et al. 2015 [74]							
De Souza et al. 2013 [89]			VBE protective ^7^ CS protective ^1,4,5^				
Di Dedda et al. 2017 [81]						CS protective ^1,7^	
Moghadam et al. 2019 [71]			CS protective ^1,3^				
Rogers et al. 2012 [67]						VB protective ^1^	
Saydam et al. 2019 [63]							
Song et al. 2014 [79]				AVB protective ^8^	CS protective ^2,3,4^	CS protective ^2,3,4^	
Spaich et al. 2020 [78]							
Yee et al. 2013 [68]							
Zgliczynska et al. 2021 [77]	SVB protective ^2^	SVB protective ^2^				CS protective ^2^	
	Adjusted association between mode of birth and sexual function						
	Crude association between mode of birth and sexual function						
	No association between mode of birth and sexual function						
	No association calculated between mode of birth and sexual function						

SVB = spontaneous vaginal birth, VBE = vaginal birth with episiotomy, AVB = assisted vaginal birth, pCS = planned caesarean section, eCS = emergency caesarean section, VB = vaginal birth (without differentiation between SVB, VBE and AVB), CS = caesarean section (without differentiation between pCS and eCS), ^a^ No differentiation between VBE and AVB, ^b^ No differentiation between SVB and VBE; Domains: ^1^ Overall sexual dysfunction, ^2^ Arousal, ^3^ Desire, ^4^ Lubrication, ^5^ Orgasm, ^7^ Pain, ^8^ Satisfaction with sexual life with partner.

**Table 6 ijerph-20-05252-t006:** Longer-term associations between combinations of modes of birth and sexual function (≥ 12 months postpartum).

	**SVB vs. CS**	**VBE vs. CS**	**AVB vs. VB ^a^**	**AVB vs. CS**	**VB vs. CS**
Baytur et al. 2005 [59]					
Cai et al. 2014 [84]					
Chang et al. 2015 [85]					
Dahlgren et al. 2022 [86]					
De Souza et al. 2015 [74]					
De Souza et al. 2013 [89]					
Dean et al. 2008 [73]	CS protective ^8^			CS protective ^8^	
Fehniger et al. 2013 [66]			VB protective ^3^	CS protective ^3^	
Ghorat et al. 2017 [70]					CS protective ^2^
Gungor et al. 2007 [61]					
Moghadam et al. 2019 [71]		CS protective ^3^			
Spaich et al. 2020 [78]					
	Adjusted association between mode of birth and sexual function				
	Crude association between mode of birth and sexual function				
	No association between mode of birth and sexual function				
	No association calculated between mode of birth and sexual function				

SVB = spontaneous vaginal birth, VBE = vaginal birth with episiotomy, AVB = assisted vaginal birth, pCS = planned caesarean section, eCS = emergency caesarean section, VB = vaginal birth (without differentiation between SVB, VBE and AVB), CS = caesarean section (without differentiation between pCS and eCS), ^a^ No differentiation between SVB and VBE; Domains: ^2^ Arousal, ^3^ Desire, ^8^ Satisfaction with vaginal tone.

## Data Availability

Not applicable.

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
