# Peer review of "The Impact of Mode of Birth, and Episiotomy, on Postpartum Sexual Function in the Medium- and Longer-Term: An Integrative Systematic Review"

_ijerph, 2023, doi:10.3390/ijerph20075252_

Round 1

Reviewer 1 Report

This has the potential to be an interesting review but more structured approach is needed for reporting both the study characteristics and the review findings for readers to be able to navigate successfully. I have made a number of suggestions for you to consider. These are not listed in order of importance.

(1) Suggest that longer-term is a better descriptor of studies looking at impact after 12 months than long-term.

(2) The stated aim of this review is “to examine the impact of different modes of birth, as well as episiotomy, on sexual functioning, beyond six months postpartum in the medium-term (≥ 6 months and < 12 months postpartum) and long-term (≥ 12 months postpartum).” (Lines 93-95) but this does link to the earlier statement that, “An important aspect that may impact sexual function is mode of birth, which can be categorized into spontaneous vaginal birth, vaginal birth with an episiotomy, assisted vaginal birth, planned caesarean section or emergency caesarean section” (Lines 40-41). Despite the implication that the lack of information about these different categories of mode of birth (MoB) is a significant reason that the review was undertaken, there is no explicit use of this categorisation in the review, I suggest that by making use of these five MoB categories, would provide a useful structure for the review, and possibly, by doing so, encourage its use in future studies. The five MoB categories would lead to 10 direct comparisons, that should be the main focus of the review:

1.       SVB vs VBE

2.       SVB vs AVB

3.       SVB vs pCS

4.       SVB vs eCS

5.       VBE vs AVB

6.       VBE vs pCS

7.       VBE vs eCS

8.       AVB vs pCS

9.       AVB vs eCS

10.   pCS vs eCS

Other comparisons involve the collapsing of two or more groups, and could be a secondary focus of the review (in some cases these may be only data available, at this stage):

1.       SVB vs VB (includes VBE and AVB)

2.       SVB vs CS (includes pCS and eCS)

3.       VBE vs CS (includes pCS and eCS)

4.       AVB vs VB (includes SVB and VBE)

5.       AVB vs CS (includes pCS and eCS)

6.       VB (includes SVB, VBE and AVB) vs CS (includes pCS and eCS)

7.       VB (includes SVB, VBE and AVB) vs pCS

If the authors decide to adopt this approach, it would need to be explicitly stated in the methods section and Tables 3a and 3b restructured to reflect this.

(3) Tables 3 would be vastly improved by changing the axes, so that the outcomes are reported on the x-axis.

(4) The paragraph in the Introduction outlining other systematic reviews should clearly justify the need for this review. I suggest that this paragraph is revised to give an overall summary of the findings of other reviews and greater focus on the methodological shortcomings of the existing reviews, each shortcoming could then be addressed in the methods section of this review. (The methods would also include reference to the primary focus of analysis being the five categories of mode of birth, and subsequent comparisons a secondary focus, if this suggestion is accepted).

(5) It would be useful to include some information about potential confounders in the introduction, as these justify aspects of data extraction, rather than bringing these up later.

(6) There is reference to “other studies” but only one study cited (Line 65). Where possible it can be helpful to refer to the study type, in this case it is a cohort study.

(7) Under 2.2. Inclusion and exclusion criteria (Line 106) the authors state that “Both quantitative and qualitative studies were eligible for inclusion in this review, including cohort studies, cross-sectional studies, case-control studies and randomised controlled trials” (Lines 107-108). I think the authors mean “mixed method” studies rather than “qualitative studies”, as “only studies that used a validated sexual function measures were included (Line 117) is unlikely in a qualitative study.

(8) In Section 2.3, Valid measures of sexual function (Line 119) six validated measures are summarised but there is no mention of the “gold standard” later referred to in Section 2.6 Quality assessment, where the authors state ““We adapted the NOS, as self-reported measures of sexual function was considered the gold standard for assessing sexual measures by the authors (my emphasis), since sexual function can only be reported by the person experiencing it.” (Lines 189-191). This should be reconciled by including self-reported measures in the earlier section.

(9) Under 3.1. Search (Line 213), “In total, 3022 studies were identified in database searches” (Line 214) yet the flowchart shows 6509. There is another error in the flowchart: Records screened on title= 5739 and Excluded =5632, yet this leaves 107 studies not 112.

(10) Table 1: Please left justify text in tables rather than centre it. It makes it much easier to read. I also suggest sorting the studies by study design: cross-sectional studies, case-control studies, retrospective cohort studies, then prospective cohort studies. Again, this makes it easier for readers to compare different studies. In this table, I suggest that something like “Population characteristics” rather than “Race” may be a better field, which will also provide a place to report the parity of the study population.

(11) It is not necessary to say “A total of” when reporting the number of studies – just state the number, and focus on the outcome. If my suggestion of making the primary focus the five mode of birth categorisations is accepted, these should be reported in the same order as listed in the tables.

(12) It is helpful to include the total number of women in each group of studies when reporting the study characteristics (Lines 224-226), for example:14 prospective cohort studies (x women; including at least y primiparous women), nine cross-sectional studies (x; y), three case-control (x; y) studies and five retrospective cohort studies (x;y). Including detail like this gives readers better understanding of the evidence base.

(13) In regard to Section 3.6. Overall (Line 340) start with a summary sentence of the factors that are considered potentially important: study design, sample size, parity, adjustment for confounders; timing of follow-up; and study quality.

(14) An important limitation for this review is that there is no consensus on reporting standards, including reporting for mode of delivery and adjustments for confounding. This is missing.

Author Response

Dear Reviewer, 

We would like to express our sincere gratitude to you for your meticulous review of our manuscript. We believe that your comments, and the changes we have made to the manuscript in response to them, have substantially improved the manuscript. 

Please see attached a point-by-point response to your comments as a word document. In italics is our response to your comment. In bold you can see the section that has been changed as a result of your comment (along with the line number this change occurs on), and underneath the bold is the change that has been made in the manuscript. 

We thank you again for your review.

Kind regards

Reviewer 2 Report

Clearly a lot of work has gone into this manuscript. Please find my comments in the attached document.

Author Response

(The authors gave the same response as above.)

Round 2

Reviewer 1 Report

Thank you for responding to the earlier feedback so thoroughly.

Reviewer 2 Report

Thank you for attending to my previous comments in great detail. 

Congratulations on a detailed and worthwhile project.